

# Mixing characteristics of refractory black carbon aerosols determined by a tandem CPMA-SP2 system at an urban site in Beijing

Hang Liu[1,2], Xiaole Pan[1], Dantong Liu[3], Xiaoyong Liu[1,4], Xueshun Chen[1], Yu Tian[1], Yele Sun[1,2,4], Pingqing Fu[5], Zifa Wang[1,2,4]

[1] State Key Laboratory of Atmospheric Boundary Layer Physics and Atmospheric Chemistry, Institute of Atmospheric Physics, Chinese Academy of Sciences, Beijing, 100029, China

[2] University of Chinese Academy of Sciences, Beijing, 100049, China

[3] Department of Atmospheric Sciences, School of Earth Sciences, Zhejiang University, Hangzhou, Zhejiang, 310027, China

[4] Center for Excellence in Regional Atmospheric Environment, Chinese Academy of Science, Xiamen, 361021, China

[5] Institute of Surface-Earth System Science, Tianjin University, Tianjin 300072, China

Correspondence to: Xiaole PAN (panxiaole@mail.iap.ac.cn)

**Abstract** Black carbon aerosols play an important role in climate change by absorbing solar radiation and degrading visibility. In this study, the mixing state of refractory black carbon (rBC) at an urban site in Beijing was studied with a single particle soot photometer (SP2), as well as a tandem observation system with a centrifugal particle mass analyzer (CPMA) and a differential mobility analyzer (DMA), in early summer of 2018. The results demonstrated that the mass-equivalent size distribution of rBC exhibited an approximately lognormal distribution with a mass median diameter (MMD) of 171.2 nm. When the site experienced prevailing southerly winds, the MMD of rBC increased notably by 19%. During the observational period, the ratio of the diameter of rBC-containing particles ($D_p$) to the rBC core ($D_c$) was 1.20 on average for $D_c$=180 nm, indicating that the majority of rBC particles were thinly coated. The $D_p/D_c$ value exhibited a clear diurnal pattern, with a maximum at 1400 LST and an enhancing rate of 0.013/h; higher Ox conditions increased the coating enhancing rate. Bare rBC particles were primarily in a fractal structure with a mass fractal dimension ($D_{fm}$) of 2.35, with limited variation during both clean and pollution periods, indicating significant impacts from on-road vehicle emissions. The morphology of rBC-containing particles varied with aging processes. The mixing state of rBC particles could be indicated by the mass ratio of non-refractory matter to rBC ($M_R$). In the present study, rBC-containing particles were primarily found in an external fractal structure when $M_R < 1.5$ and changed to a core-shell structure when $M_R > 6$, at which the measured scattering cross section of rBC-containing particles was consistent with that based on the Mie-scattering simulation. We found only 9% of the rBC-containing particles were in core-shell structures on clean days with a particle mass of 10 fg, and the number fraction of core-shell structures increased considerably to 32% on pollution days. Considering the morphology change, the absorption enhancement ($E_{abs}$) was 11.7% higher based on core-shell structures. This study highlights the combined effects of morphology and coating thickness on the $E_{abs}$ of rBC-containing particles, which will be helpful for determining the climatic effects of BC.





## 1 Introduction

Black carbon (BC) aerosol is one of the principal light-absorbing aerosols in the atmosphere. It is regarded as the second most
important component contributing to global warming only after $CO_2$ (Jacobson, 2000). BC has a much shorter lifetime than
$CO_2$. Thus, BC's radiative perturbation on a regional scale may be distinct from globally averaged estimates. It has been
reported that BC's direct radiative forcing can reach an order of $+10$ W m$^{-2}$ over East and South Asia (Bond et al., 2013). BC
aerosol can also influence the climate by altering cloud properties, such as the evaporation of cloud droplets, reduction in cloud
lifetime and albedo (Ramanathan et al., 2001;Ramanathan and Carmichael, 2008). Ding et al. (2016) determined the existence
of BC in the upper mixing layer could absorb downward solar radiation, impeding the development of the boundary layer,
which aggravates air pollution. Moreover, BC aerosols have detrimental health effects. BC and organic carbon are regarded
as the most toxic pollutants in PM$_{2.5}$, possibly leading to ~3 million premature deaths worldwide (Apte et al., 2015;Lelieveld
et al., 2015).

BC is typically emitted from the incomplete combustion of fossil fuels and biomass. After being emitted into atmosphere, BC
particles tend to mix with other substances through coagulation, condensation and other photochemical process, which
significantly changes BC's cloud condensation nuclei activity as well as its light absorption ability (Liu et al., 2013;Bond and
Bergstrom, 2006). Model results suggest that after BC's core is surrounded by a well-mixed shell, its direct absorption radiative
forcing could be 50% higher than that of BC in an external mixing structure (Jacobson, 2001). Such an absorption enhancement
phenomenon is interpreted as exhibiting a "lensing effect", in which a non-absorbing coating causes more radiation to interact
with the BC core and thus more light is absorbed. This absorption enhancement effect has been proven in laboratory studies
(Schnaiter et al., 2005). Shiraiwa et al. (2010) reported that the absorption enhancement of BC in a core-shell structure
increased with coating thickness and can reach a factor as high as 2. Nevertheless, field observation results demonstrated large
discrepancies (6 to 40%) in the absorption enhancement of aged BC particles (Cappa et al., 2012;Lack et al., 2012). The
discrepancies could be attributed to the complex mixing state of BC in the real atmosphere, which depends on coating
composition, coating amount as well as the size of the BC core and structure. Bond et al. (2013) regarded the mixing state of
BC as being one of the most important uncertainties in evaluating BC direct radiative forcing. Furthermore, freshly emitted
BC is initially hydrophobic. Mixing BC with other soluble materials will significantly increase BC-containing particles'
hygroscopicity and thus the ability to become cloud condensation nuclei (Zhang et al., 2008;Popovicheva et al., 2011). This
ability is associated with the wet deposition rate and consequently influences the lifetime and spatial distribution of BC
particles in the atmosphere. For these reasons, more observations are needed to determine the specific spatial and temporal
distribution of BC's mixing state, which would be helpful for minimizing the uncertainty in evaluating BC's climatic and
environmental effects.



China's economy has grown rapidly in recent decades, accompanied by the substantial emission of pollutant precursors. Annual emissions of BC in China are reported to have increased from 0.87 Tg in 1980 to 1.88 Tg in 2009, comprising half of the total

emissions in Asia and an average of 18.97% of the global BC emissions during this period (Qin and Xie, 2012). Such substantial BC emissions greatly influence the regional climate and environment (Menon et al., 2002;Ding et al., 2016). Although temporal/spatial variations of BC and corresponding optical properties of aged BC have been recently reported (Cao et al., 2004;Cao et al., 2007;Zhang et al., 2009), the observational studies on BC's mixing state remain insufficient. Recently, the Single Particle Soot Photometer (SP2) has been used as a reliable instrument for estimating the mixing state of BC due to

its single particle resolution and high accuracy. Several studies have used SP2 to investigate BC's mixing state in China (Huang et al., 2012;Gong et al., 2016;Wu et al., 2016;Wang et al., 2014;Yang et al., 2018). Most studies have primarily focused on the variability of BC's mixing state on severe haze days during winter because of the extremely high concentrations of particle matter and low visibility. Local emissions and the long-range transport of BC have been discussed (Gong et al., 2016;Zhang et al., 2018). In summer, longer and higher radiation and high hydroxyl radical concentration favors photochemical reaction

and thus contributes to the condensation aging of BC. By using a smog chamber, Peng et al. (2016) found that BC-containing particles experienced a fast increase owing to the photochemical aging of BC coating materials in Beijing's urban environment, even in relatively clean conditions. Cheng et al. (2013) noted that the changing rate of BC from an external to internal mixing state can reach up to 20%/h in summer. Thus, the mixing state of BC should also be carefully considered for the relatively clean days during summer.

In this study, we used a SP2 to investigate BC in the urban areas of Beijing, China, during early summer, focusing on the size distribution and mixing state of BC-containing particles. A tandem experiment combining a centrifugal particle mass analyzer (CPMA, Cambustion Ltd.) and a differential mobility analyzer (DMA, model 3085A, TSI Inc., USA) with a SP2 were performed during two typical cases, and the microphysical properties of the BC-containing particles were analyzed in the discussion section, focusing on BC's morphology and light absorption enhancement for freshly emitted BC in urban areas.

Various techniques have been developed to quantify the mass concentration of BC aerosols including optical, thermal, thermal-optical or photoacoustic methods. At present, the mass concentration of BC was measured on the basis of incandescent signal emissions, therefore, refractory black carbon (rBC) was used. We follow this rule in the following sections. The abbreviations and symbols used in this paper are listed in Table S1.

## 2 Observation and methodology

### 2.1 Site description

The measurement of rBC particles was performed from May 30 to June 13, 2018, in an air-conditioned container located in the tower campus of the State Key Laboratory of Atmospheric Boundary Layer Physics and Atmospheric Chemistry, Institute of Atmospheric Physics (LAPC, longitude: 116.37°E; latitude: 39.97°N). The sampling site is located between the northern third and fourth ring roads of Beijing, approximately 50 m from the closest road and 380 m away from the nearest Jingzang





highway (Fig. 1). Anthropogenic emissions in the observational campus were limited. This site was well representative of the urban conditions in Beijing.

An aerosol sampling inlet was positioned 4 m above the ground. A PM$_{2.5}$ cyclone (URG-2000-30ENS-1) was used to selectively measure particles with an aerodynamic diameter smaller than 2.5 µm, because rBC particles are typically present in the submicron mode. The systematic configuration of rBC measurements is presented in Fig. 2a. A supporting pump with

a flow rate of 9.6 L/min was used to guarantee a total inlet flow rate of 10 L/min (the demanding flow rate of a PM$_{2.5}$ cyclone) and to minimize particle loss in the tube. The residence time of the sampling flow was estimated to be ~17 s. Theoretical calculation revealed the maximum Reynolds number in the measurement system reached 1,197, implying that air flow in the tube was generally laminar. The SP2 was operated at a sampling flow rate of 100 ccm. The data acquisition of SP2 was set to collect one of every ten particles. The SP2 laser intensity was approximately constant by performing a polystyrene sphere latex

(PSL) calibration before and after the campaign (Fig. S1)

.

## 2.2 Instruments

### 2.2.1 Single-particle soot photometer (SP2)

A single particle soot photometer (SP2, Droplet Measurement Technology, Inc., Boulder, CO, USA) was used to determine

the mass size distribution and mixing state of rBC particles in the atmosphere. In the SP2 measuring chamber, an intensive continuous intracavity Nd:YAG laser beam is generated (1064 nm, TEM00 mode). After a rBC-containing particle crosses the beam, it is heated to its boiling point (3500–4000 K) by sequentially absorbing the laser power and emitting incandescent light. Maximum incandescence intensity (or the peak height of the incandescence signal) is approximately linearly correlated with rBC's mass, irrespective of the presence of non-BC material or rBC's morphology. The SP2 also records the scattering profile

of rBC-containing particles passing through the laser beam, which could be useful for estimating the aerosol type and mixing state. The pure scattering particles can also be distinguished if they only produce scattering signals without incandescence signals. The SP2's detection range of rBC is 70–600 nm and the uncertainty of the derived rBC mass is 31% (Shiraiwa et al., 2008).

### 2.2.2 Centrifugal particle mass analyzer (CPMA)

A centrifugal particle mass analyzer can select aerosol particles with known mass on the basis of a specific charge-to-mass ratio (Fig. S2). By applying an electric field and rotating, the CPMA imposes opposite centrifugal and electric forces to the charged aerosols inside. Aerosols will successfully pass through CPMA only when these two forces are balanced, as presented by Eq. (1):

$$mr\omega^2 = \frac{N_q eV}{r ln(^{r_0}/_{r_i})} \tag{1}$$




where e is the electronic charge ($1.6 \times 10^{-19}$ C), V is the voltage inside the CPMA, $r_0$ and $r_i$ are the outer and inner cylinders' radii, and r is the center radius between the two cylinders. Superior to an aerosol particle mass analyzer (APM), the cylinders of the CPMA adopt a different angular velocity, which improves the CPMA's stability and transfer function (Olfert and Collings, 2005).

**2.3 Data analysis**

**2.3.1 Mass-equivalent size distribution of rBC particles**

Before measurement, the SP2 was calibrated to determine the relationship between the incandescence peak height and the mass of rBC particles using Aquadag aerosols (Acheson Inc., USA). Fig. 2b illustrates the schematic diagram of the calibration system. During calibration, monodisperse Aquadag aerosols were generated with an atomizer (model 3072, TSI Inc., USA) and dried using a diffusion dryer. Then, Aquadag aerosols with known mass ($M_{rBC}$) were selected with a CPMA and were injected into the SP2 to obtain the corresponding laser-induced incandescence (LII) signal. The LII-$M_{rBC}$ relationship is thus obtained (Fig. S3). A calibration procedure was also performed by combining the DMA and SP2. Fig. S4 presents the LII-$M_{rBC}$ relationship derived from the two calibration procedures, which are approximately the same. A recent study (Laborde et al., 2012) demonstrated that the mass of rBC particles could be underestimated using Aquadag aerosol as the calibration material. We performed a correction dividing by a factor of 0.75 for each set of $M_{rBC}$ points during the calibration (Zhang et al., 2018;Liu et al., 2014). Assuming a void-free, ideally spherical structure and 1.8 g cm$^{-3}$ density for rBC material density (Bond et al., 2013), the rBC mass can be transformed to a mass equivalent diameter (MED), according to the following equation:

$$D_c = \sqrt[3]{\frac{6*M_{rBC}}{\pi*\rho_{rBC}}} \tag{2}$$

The SP2's laser current was held at a constant value of 1750 mA through the entire investigation. During calibration, the number concentration of particles was concurrently measured by both the SP2 and a condensation particle counter (CPC, model 3775, TSI Inc., USA), and a detection efficiency of rBC was determined. For large particles, the SP2's detection efficiency was approximately unified and decreased gradually for smaller rBC particles (Fig. S5). In this study, the SP2's low detection bound was set to $D_c = 70$ nm. The mass concentrations of rBC may be underestimated because of the low detection efficiency of smaller rBC particles. By extrapolating a lognormal function fit to the observed mass distribution, we found that rBC particles without the detection range caused an ~15% underestimation of the rBC mass concentration. To compensate, the mass concentration of rBC was corrected by multiplying by a factor of 1.17 during the measurement.

**2.3.2 Scattering cross section of rBC-containing particles**

The SP2 consists of two types of detectors to receive a particle's scattering signal. One detector records the entire scattering profile of the particles. The other, a two-elemental avalanche photodetector (TEAPD), is used to obtain the actual position of a particle in the intracavity. The scattering signal of the SP2 was calibrated using polystyrene latex spheres (PSL, Nanosphere





Size Standards, Duke Scientific Corp., USA) with known sizes (203±3 nm: Lot #185856; 303±3 nm: Lot #189903; 400±3 nm: Lot #189904). For purely scattering particles, the peak intensity of the scattering signal can be converted to its scattering cross section by dividing the laser intensity. For rBC-containing particles, the scattering cross section significantly decreased owing

to the evaporation of the nonrefractory coating as the rBC core absorbed energy. In the present study, the LEO-fit method was used to reconstruct an undisturbed scattering signal (Gao et al., 2007), and the data before a length that is 2.5σ (σ denotes the standard deviation of the Gaussian function) away from the beam center was used for Gaussian fitting (Pan et al., 2017). A stricter criterion (e.g., >3σ) would increase the risk of failure of fitting due to high noise, and a looser criterion (e.g., < 2σ) would increase the risk of the onset of coating evaporation. A more detailed and graphic description of the LEO-fitting method

has previously been described (Laborde et al., 2012;Liu et al., 2014). The peak intensity of the fitted scattering signal can be converted to the scattering cross section of rBC-containing particles in a similar fashion. This scattering cross section is directly measured by the SP2 and is referred to as $SCS_{sp2}$ in the following section.

For a tandem CPMA-SP2 system, the mass of a rBC-containing particle ($M_p$) may be directly selected. By assuming a core-shell structure, the diameter of the rBC-containing particle ($D_p$) may be calculated by solving the equation:

$$M_p = \frac{\pi}{6}\left(D_p^3 - D_c^3\right) * \rho_{coat} + \frac{\pi}{6}D_c^3 * \rho_{rBC} \qquad (3)$$

The coating density was set to 1.5 g/cm³ (Qiao et al., 2018) in this study. Therefore, the scattering cross section of a rBC-containing particle may be calculated using the Mie scattering theory, providing the refractive indices of the rBC core and non-rBC shell materials. The selected refractive indices were considered in this study (Fig. S6). The most proper refractive indices are determined to be 2.26-1.26i and 1.48-0i for rBC and coating material. The scattering cross section calculated in

this manner is referred to as $SCS_{mie}$ in the following section.

### 2.3.3 Calculation of the coating thickness of the rBC core

For a single SP2 measurement, $SCS_{sp2}$ was used to estimate the diameter of a rBC-containing particle ($D_p$) based on the Mie scattering theory under the assumption of a core-shell structure. $D_p/D_c$ is used to represent the coating thickness of rBC from measurements of optical size of coated BC-containing particle.

For measurements combining SP2 and CPMA, the mass ratio ($M_R$) of the coating material ($M_{non-rBC}$) and rBC ($M_{rBC}$) was also used to represent the coating thickness of rBC. The $M_R$ is calculated using the following equation:

$$M_R = (M_p - M_{rBC})/M_{rBC} \qquad (4)$$

### 2.3.4 Effective density of rBC

The effective density of bare rBC can be obtained through a tandem DMA-SP2 system, as $M_{rBC}$ and $D_{mob}$ can be measured

simultaneously. Bare rBC is defined as rBC with $D_p/D_c \approx 1$, and the effective density of bare rBC was calculated according to the equation:

$$\rho_{eff} = \frac{6M_{rBC}}{\pi D_{mob}^3} \qquad (5)$$





In principle, the measured effective density is the same as the material density if the particle features an ideal spherical shape with no void space. Thus, the effective density is an indicator of particle compactness by comparing the effective density and
material density. Several studies that include the coupling of DMA with APM or CPMA have been conducted to determine the $\rho_{eff}$-$D_{mob}$ relationship of the laboratory rBC sample Aquadag (Gysel et al., 2011;Moteki and Kondo, 2010). The relationship between $\rho_{eff}$ and $D_{mob}$ of Aquadag is presented in Fig. S7. The $\rho_{eff}$ obtained using the DMA-SP2 system in this study agreed well with previous research.

### 2.3.5 Determination of absorption enhancement

The absorption enhancement ($E_{abs}$) of a single rBC-containing particle was calculated as the ratio of the absorption cross section of a rBC-containing particle ($C_{abs,p}$) and the absorption cross section of a rBC core ($C_{abs,rBC}$) using the Mie theory with a core-shell model.

$$E_{abs} = \frac{C_{abs,p}}{C_{abs,rBC}} \qquad (6)$$

## 3 Results

### 3.1 Concentrations of PM$_{2.5}$, rBC and pollutant gases

The temporal variations of mass concentrations of PM$_{2.5}$, rBC and gaseous pollutants (O$_3$, NO$_2$) during the project are presented in Fig. 3. The mass concentration of PM$_{2.5}$ ranged between 5 and 120 μg/m$^3$ on a daily basis during the observation period. Mixing ratios of both NO$_2$ and O$_3$ exhibited obvious diurnal variations with maximum values of 68 and 145 ppbv, respectively.
O$_3$ dominant pollution occurred at 1400 LST on June 2, with a maximum of 145 ppbv, reflecting high atmospheric oxidant levels and strong photochemistry during the observation. The mass concentration of rBC was 1.21 ± 0.73 μg/m$^3$ on average, accounting for 3.5 ± 2.4% of PM$_{2.5}$ on an hourly basis. This was comparable to the previous filter-based measurement in Beijing, with an average fraction of 3.2% in the summer of 2010 (Zhang et al., 2013). The mass concentration of BC measured using a multiple-angle absorption photometer (MAAP) that correlated well (r$^2$ = 0.8) with the mass concentration of rBC
derived from the SP2 measurement; however, the optical method using the MAAP was 40% higher than that using the SP2 (Fig. S8). For the MAAP, the mass concentration of BC was converted from the actual measured absorption coefficient by assuming a constant mass absorption cross-section of BC. This assumption may cause overestimation as BC's mass absorption cross-section may be affected by coating (Tasoglou et al., 2018;Slowik et al., 2007). During the period June 1 to 6, the meteorological condition was characterized by low relative humidity (RH < 40%) and strong solar radiation, which was
favorable for ozone formation. Ambient RH increased up to 80% after June 9. Under such conditions, PM$_{2.5}$ experienced steady growth, increasing from 10 to 120 μg/m$^3$. Three rainfall events occurred on June 4, 7 and 9 (Fig. 3d). After each rainy period, most of the major pollutants decreased due to significant wet scavenging. A heavy rainfall event occurred from 0300–0700



LST on June 7, during which time the mass concentration of PM$_{2.5}$ decreased from 65 to 10 μg/m$^3$ and the mass concentration of rBC decreased from 2.63 to 0.2 μg/m$^3$.

Footprint analysis (Fig. S9) shows that Beijing was largely affected by the local air mass during the period June 7–9 due to a weak wind speed. We focused on two episodes. The first episode is the clean period on June 8, when the concentration of PM$_{2.5}$ and O$_3$ averaged 20 μg/m$^3$ and 60 ppbv, respectively, after the heavy rainy period. The second episode is the pollution episode on June 13 when the hourly mass concentration of PM$_{2.5}$ exceeded 110 μg/m$^3$. The tandem CPMA-DMA-SP2 experiment was conducted on June 8 and 13 to study the detailed characteristics of freshly emitted rBC and aged rBC particles.

**3.2 Size distribution of rBC**

The number and mass distribution as a function of the rBC's MED are illustrated in Fig. 4. As presented, the mass median diameter (MMD, the MED at the peak of the mass distribution) was 171.2 nm during the project. A brief summary of the SP2's observations in China is presented in Table 1. Most previous studies focused on the rBC's characteristics in winter, when a larger MMD (~200–230 nm) was obtained (Huang et al., 2012;Wang et al., 2014;Gong et al., 2016;Wu et al., 2017b) than in

this study. A similar MMD (180 nm) was reported in urban Shenzhen during a summer observation period (Lan et al., 2013), and a higher MMD (210–222 nm) was reported in winter. Liu et al. (2014) also found a winter-high-summer-low trend for rBC sizes in London, with MED=149±22 nm in winter and 120±6 nm in summer. Laboratory studies have proven MMD is highly dependent on combustion conditions (Pan et al., 2017) and material. Thus, MMD is a suitable indicator for the source of rBC. Several studies suggest that the MMD of rBC from biomass burning and coal is much larger than that from traffic

emissions (Wang et al., 2016;Schwarz et al., 2008). Huang et al. (2012) found the MMD observed at rural sites to be much larger than that observed at urban sites because urban sites are primarily affected by rBC emitted from traffic sources and rural sites are more influenced by rBC from coal combustion. The seasonal trend of MMD may be partially explained by the different rBC sources in summer and winter.

Fig. 5 exhibits the temporal variations in mass size distribution of rBC during the entire investigation period. Most rBC

particles were within the size range of 70–300 nm with a clear diurnal pattern. The diurnal cycle reached a peak plateau between 0300–0700 LST and it decreased gradually in the afternoon, which was controlled by the combination effects of the development of a planetary boundary layer (PBL) variation and on-road rBC emissions.

Significant change of rBC's mass size distribution occurred on June 7, corresponding to the heavy rain period. After the heavy rain event, the MMD decreased sharply to 159 nm. This decreasing trend of MMD also occurred on June 4 during another

rainfall event. Taylor et al. (2014) observed that the rBC core size distribution shifted to smaller sizes after a biomass burning plume passed through a precipitating cloud, attributing this shift to the preference of nucleation scavenging to larger rBC cores. By counting the MMD on non-rain days and rain days, (Wang et al., 2016) also found the MMD decreased from 164±21 nm to 145±25 nm.

After a rain day, on June 5, the MMD increased significantly up to 190 nm. However, the MMD was relatively stagnant and

measured approximately 160 nm during the period June 8–9. Changes in wind direction might partially explain the discrepancy



between the observations on June 5 and June 8–9. A pollutant rose plot of MMD versus wind speed and wind direction is presented in Fig. 6a. The MMD of rBC was ~160 nm at a low wind speed condition and exhibited a significant increase with an increasing southeast wind speed. The maximum MMD exceeded 190 nm when the wind speed was greater than 10 m/s. Fig. 6b presents the correlation between windspeed and MMD. A south wind period was selected when the wind direction was

135–225° and a north wind period was the time when the wind direction was 325–45°. The MMD exhibited little correlation with wind speed and varied little between the south and north wind periods when the wind speeds were less than 2 m/s, as local nascent rBC emissions were predominant. A MMD of 150–160 nm at low wind speed time may be characteristic of local sources. The MMD had a strong positive correlation with the wind speed in the south wind period ($r^2$ =0.93) suggesting the rBC from the south was larger.


### 3.3 Temporal variation of $D_p/D_c$

The $D_p/D_c$ for a given single rBC-containing particle is calculated using the LEO fitting method. Herein, rBC cores with $D_c$=180 ± 10 nm were selected because the low scattering signal of a small size rBC was more easily influenced by signal noise ($D_p/D_c$ indicates the $D_p/D_c$ with $D_c$=180 ± 10 nm in the following discussion if not specified). The $D_p/D_c$ variation during

the observation time is illustrated in Fig. 7. In general, $D_p/D_c$ was 1.20±0.05 on average during the investigation, which is consistent with the observation (1.15) during the summer in Paris (Laborde et al., 2013). rBC sources and the aging process significantly influence the $D_p/D_c$ of rBC. The rBC from traffic is reported to be relatively uncoated (Liu et al., 2017), whereas the rBC emitted by biomass burning is found to be moderately coated with $D_p/D_c$=1.2–1.4 (Pan et al., 2017). Moreover, $D_p/D_c$ increases with the aging process and a larger $D_p/D_c$ (1.6) was found in an aged continental air mass (Shiraiwa et al., 2008).

The relatively low $D_p/D_c$ value further supports the argument that rBC was primarily emitted from on-road vehicles during the summer in Beijing.

The $D_p/D_c$ distributions for the two episodes before the tandem CPMA-DMA-SP2 experiments are shown in Fig. 7. Episode 1 occurred after a heavy rain period and is representative of a clean condition. Episode 2 is characterized with the highest $D_p/D_c$ value (1.4) and the highest PM$_{2.5}$ concentration value (120 μg/m$^3$). During episode 1, the $D_p/D_c$ distribution exhibited a

single peak at 1.05. However, during episode 2, two $D_p/D_c$ distribution peaks were found. The PM$_{2.5}$ and NO$_2$ concentrations experienced a step increase in episode 2, as depicted in Fig. 3. Such a steep increase was often regarded as the regional transport of pollution (Wu et al., 2017a;Li et al., 2017). 63% of the rBC was estimated to be transported from outside of Beijing during the pollution event and the rBC-containing particles from regional transportation are characterized by more coating material and more absorption ability (Zhang et al., 2018). The steep increase in pollutants and two modes of $D_p/D_c$ indicate regional

transportation may significantly influence the mixing state of rBC in the pollution cases.



## 3.4 Diurnal variation of $D_p/D_c$

The temporal variation of $D_p/D_c$ exhibited a clear day-high and night-low pattern. Fig. 8 exhibits the diurnal trend of $D_p/D_c$. The mean $D_p/D_c$ increased during the daytime with a peak (1.2) at 1400 LST and a minimum (1.12) at 0600 LST. $D_p/D_c$ was controlled by the competing effect of emissions and aging, because freshly emitted thinly coated rBC tended to decrease $D_p/D_c$ and the aging process tends to increase $D_p/D_c$. The increasing trend of $D_p/D_c$ during the day could be explained by the prevailing aging process, whereas the decreasing trend at night can be explained by the prevailing emission process, as the photochemical condensation aging during the day was much faster than the coagulation aging at night (Riemer et al., 2004;Chen et al., 2017). By counting the $D_p/D_c$ from 0600–1400 LST, the $D_p/D_c$ enhancing rate was calculated to be 0.013/h, with $D_c$=180 nm corresponding to a 2.34 nm/h enhancing rate of $D_p$. Lager $D_p/D_c$ enhancing rate was found in period with high $O_x$ concentration which may be favorable to the formation of coating material upon rBC. The photochemical process and condensation aging have proven to be very efficient during the day. Using a smog chamber, Peng et al. (2016) found the $D_p$ enhancing rate of rBC-containing particles could reach 26 nm/h in Beijing's urban area. Although the photochemical process and condensation may rapidly increase the $D_p$, the difference between the present study and the smog chamber results indicated that the "apparent" $D_p/D_c$ enhancing rate in the ambient measurement was relatively small given the continuous freshly emitted rBC in urban Beijing. Thus, the $D_p/D_c$ was always at a low level, resulting in little light absorption enhancement during the summer.

## 4 Discussions

### 4.1 Morphological evolution of rBC-containing particles

#### 4.1.1 Morphology of bare rBC

By coupling DMA and SP2, the mass and mobility diameter of bare rBC ($D_p/D_c \approx 1$) can be simultaneously obtained and, therefore, the effective density ($\rho_{eff}$) can be calculated. The $\rho_{eff}$ of the ambient bare rBC was measured during a clean day and pollution day. The $\rho_{eff}$ of bare rBC at 200–300 nm ranged from 0.41–0.29 g/cm$^3$, which was much smaller than the material density of rBC (1.8 g/cm$^3$) (Bond et al., 2013). This significant discrepancy indicates bare rBC was in a fractal structure consistent with the electron microscopic image results that bare rBC was in a fractal chain-like structure (Li et al., 2003). $\rho_{eff}$ had no evident difference between pollution days and clean days because bare rBC particles were freshly emitted and only affected by local sources. A power law was used to describe the fractal-like aggregates of particles: $M \propto D_{mob}{}^{Dfm}$, where $D_{fm}$ is defined as the mass fractal dimension that is an indicator of particle compactness. $D_{fm}$ is 3 for ideal spherical particles and less than 3 for fractal particles. From the definition equation of $\rho_{eff}$ (Eq. 5), the following relationship exists: $\rho_{eff} \propto D_{mob}{}^{Dfm-3}$. Thus, a larger bare rBC had a smaller $\rho_{eff}$, which was consistent with the results in Fig. 9. A power function was used to fit the observation data. $\rho_{eff} \propto D_{mob}{}^{-0.65}$ and $\rho_{eff} \propto D_{mob}{}^{-0.6}$ were separately found in clean and pollution days, corresponding to the mass fractal dimensions of 2.35 and 2.4, respectively. These mass fractal dimensions from the summer in Beijing are similar to the observations ($D_{fm}$=2.3) from urban Tokyo (Moteki and Kondo, 2010) and the diesel exhaust measurement ($D_{fm}$=2.35) (Park et





al., 2004), suggesting the freshly emitted bare rBC particles originated primarily from traffic sources. Traffic may contribute
a majority of the fresh rBC during both pollution and clean periods in the summer.

### 4.1.2 Morphology of moderately-coated and thickly coated rBC

The comparison of $SCS_{sp2}$ and $SCS_{mie}$ as a function of $M_R$ for a particle mass of 10 fg is illustrated in Fig. 10a. $SCS_{sp2}/SCS_{mie}=1$ implies the scattering cross section measured by SP2 is the same as the model prediction under the assumption of a core-shell
structure. When $M_R \approx 0.1$, the rBC was considered to be in a fractal structure as discussed above. With $M_R$ increasing, $SCS_{sp2}/SCS_{mie}$ gradually decrease until $M_R=1.5$, indicating the coating material may be not sufficient to encapsulate rBC and the rBC-containing particle was not in a core-shell structure. Liu et al. (2017) showed that rBC-containing particles with $M_R<1.5$ are primarily present in an external structure. When $1.5<M_R<6$, the $SCS_{sp2}/SCS_{mie}$ steadily increased. Laboratory studies have demonstrated that soot particles will become more compact when sulfuric acid and water condense on them
(Pagels et al., 2009;Zhang et al., 2008). Ambient measurements also proved that the effective density of rBC-containing particles significantly increased from 0.5 to 1.5 g/cm³, implying a more compact structure, with an increase in coating thickness (Peng et al., 2016). These results were consistent with the $SCS_{sp2}/SCS_{mie}$ results that the shape of rBC-containing particles gradually transforms to become a more compact core-shell-like structure in this stage. When $M_R>6$, the $SCS_{sp2}/SCS_{mie}$ was equal to 1, indicating the rBC-containing particles were in a core-shell-like structure in this stage.
Similar phenomena were found in the relationship of $SCS_{sp2}/SCS_{mie}$ and $M_R$ for particle masses of 5 fg, as illustrated in Fig. 10b. However, when $M_R \approx 0.1$, the $SCS_{sp2}$ was consistent with the model prediction of $SCS_{sp2}/SCS_{mie} \approx 1$ for a particle mass of 5 fg. This is because the scattering signal was not sensitive to the irregularity of smaller-sized particles (Moteki and Kondo, 2010). Therefore, a Mie theory-based core-shell model could capture the main feature. According to the relationship between $SCS_{sp2}/SCS_{mie}$ and $M_R$, the rBC-containing particles are classified into three groups: external stage ($0<M_R<1.5$), transit stage
($1.5<M_R<6$) and internal stage ($M_R>6$). A similar variation between $SCS_{sp2}/SCS_{mie}$ and $M_R$ was found by (Wu et al., 2018;Liu et al., 2017). The transition point from transit period to internal period determined by (Liu et al., 2017) is slightly lower than that in this study, which may be indicative of the different chemo-physical properties of rBC between Beijing and London. Measurements combining CPMA and SP2 were conducted separately on a clean day (June 8) and pollution day (June 13). Fig. 11 presents the $M_R$ for different CPMA setpoints (1 fg, 2 fg, 5 fg, 10 fg) on June 8 and June 13. The median $M_R$ is 0.4 for $M_p$
=1 fg and 1.7 for $M_p$ =10 fg on the clean day, whereas the median $M_R$ is 0.7 for $M_p$ =1 fg and 3.7 for $M_p$ =10 fg on the pollution day. The median $M_R$ values of the pollution day were all larger than those on the clean day for the four $M_p$ points. This result demonstrated that rBC had more coating material during the pollution day than the clean day. The transition points between the three mixing state evolution stages were marked by gray lines in Fig. 11. The rBC-containing particles with $M_p=1$ fg and 2 fg are primarily located in the external mixing stage, regardless of pollution. With an increase in $M_p$, more rBC-containing
particles were in the transition or internal stage range. On the clean day, 7% of the rBC-containing particles were in the internal mixing state, when $M_p=10$ fg. However, on the pollution day, 32% of the rBC-containing particles were in the internal mixing





state, when $M_p$=10 fg. This phenomenon implied most rBC-containing particles are not in an ideal core-shell structure on clean days, whereas more rBC-containing particles were in a core-shell structure with thicker coating on the pollution day.

## 4.2 Implication of rBC-containing particles morphology on light absorption

### 4.2.1 Simplified absorption enhancement model

The Mie theory model was used to simulate the $E_{abs}$ of rBC under different $M_R$. As shown in Fig. 12, the $E_{abs}$ typically increased with $M_R$ increasing because of the "lensing effect", as demonstrated in previous studies (Lack et al., 2012;Nakayama et al., 2010;Schnaiter et al., 2005). An upper limitation of the "lensing effect" exists. When $M_R$>10, the $E_{abs}$ was approximately independent of $D_c$ and $M_R$, fluctuating between 1.9 and 2.2. With an increasing $M_R$, $E_{abs}$ increases more quickly and peaks earlier for rBC with a larger $D_c$. However, the results were based on the assumption of a perfect core-shell structure. As discussed previously, the morphology of rBC-containing particles changed from a chain-like fractal structure to a more compact core-shell structure with an increasing coating thickness. The mixing state of rBC-containing particles was classified into external, transition and internal state based on the $M_R$ range. The rBC-containing particles with an external mixing state were considered to have no absorption enhancement, and the rBC-containing particles with an internal mixing state were considered to have a core-shell structure and the same $E_{abs}$ of a perfect core-shell structure. The $E_{abs}$ in the transition period was calculated by the interpolation of $E_{abs}$ between the external and internal stage. The reliability of this morphology-based model has been proven by comparing the $E_{abs}$ derived from the model and measuring the $E_{abs}$ (Liu et al., 2017). A large uncertainty of $E_{abs}$ was found during the external and transition stage denoted by the pink shaded area in Fig. 12.

### 4.2.2 $D_p/D_c$ and absorption enhancement in summer

The $E_{abs}$ at 550 nm wavelength with $D_c$=180±10 nm was calculated separately using the core-shell model and morphology dependent model, as shown in Fig. 13. $E_{abs}$ was 1.15 on average using the core-shell model but only 1.03 using the new model. The $E_{abs}$ was calculated only for rBC-containing particles with $D_c$=180±10 nm. The $E_{abs}$ for the majority of rBC-containing particles might be higher because the smaller rBC in our investigation tend to be more thickly coated. $E_{abs}$ using the core-shell model overestimated by 11.7% because the observation averaged the coating thickness ($D_p/D_c$=1.2) corresponded to $M_R$=0.37, suggesting the coating material was not sufficient and most of the rBC-containing particles were not in a core-shell structure. The $D_p/D_c$ increased sharply during the PM$_{2.5}$ pollution event. This substantial increase of coating thickness changed the morphology of rBC-containing particles; more rBC-containing particles removed of the external mixing stage, resulting in a significant increase of $E_{abs}$.





## 5 Conclusion

The mixing characteristics of rBC-containing particles were investigated in Beijing during the early summer of 2018 using a single particle soot photometer (SP2). The rBC had an approximately log-normal distribution as a function of the mass equivalent diameter (MED), characterized by a mass median diameter (MMD) of 171.2 nm, which is consistent with previous urban measurements. The mass size distribution was highly associated with meteorological conditions. Heavy rain events caused the rBC mass size distribution to be smaller, indicating in-cloud nucleation scavenging may be a more efficient mechanism for rBC-containing particles. The mass size distribution of rBC shifts to larger sizes when south winds prevail, primarily caused by the different rBC sources in the south and coagulation during transport.

The $D_p/D_c$ was 1.20 on average, with $D_c$=180 nm during the investigation period, indicating a small coating thickness of rBC during the summer. $D_p/D_c$ exhibited a clear diurnal pattern with a peak at 1400 LST, increasing from 0600 to 1400 LST at a increase rate of 0.013/h, corresponding to 2.34 nm/h, with $D_c$=180 nm during the day. The increase rate was much higher in the Ox high period. However, this increase rate was significantly lower than the smog chamber results, with an increasing rate of 26 nm/h, because continuously emitted rBC lowered the $D_p/D_c$ in ambient measurements. Although photochemical aging may be very efficient, with continuously emitted rBC, the $D_p/D_c$ increase in the ambient air was very slow, indicating the rBC-containing particles were primarily at a low $D_p/D_c$ level in summer.

A tandem measuring system with a differential mobility analyzer (DMA) and a centrifugal particle mass analyzer (CPMA) were coupled with a SP2 to investigate the detailed characteristics of rBC-containing particles in summer. The results showed the effective density of bare rBC ($D_p/D_c$=1) was determined to be 0.41–0.30 g/cm$^3$ for $D_c$=200–300 nm. These effective densities were significantly lower than the rBC material density (1.8 g/cm$^3$), suggesting the bare rBC was in a fractal structure. The corresponding mass fractal dimension ($D_{fm}$) was 2.35, which agrees well with the $D_{fm}$ of the direct measurement from vehicles and was unchanged regardless of pollution, indicating traffic emissions are a major source of fresh bare rBC on both clean and pollution days during the summer in Beijing. With increasing coating thickness, the morphology of rBC changed from a fractal structure to a compact core-shell structure. When $M_R$ ($M_{coat}/M_{rBC}$) <1.5, rBC-containing particles were in an external structure. When $M_R$>6, rBC-containing particles were in a core-shell structure. When 1.5<$M_R$<6, the rBC-containing particles were in a transition stage.

Based on the core-shell model and Mie theory, a new morphologically dependent absorption enhancement ($E_{abs}$) scheme was proposed and applied in the ambient measurement. Compared with the new morphologically dependent model, the absorption enhancement ($E_{abs}$) averaged 1.03 with $D_c$=180 nm at a wavelength of 550 nm in the summer. The core-shell model overestimated $E_{abs}$ by 11.7%.



**Data availability**

To request the data given in this study, please contact Dr. Xiaole Pan at the Institute of Atmospheric Physics, Chinese Academy of Sciences, via email (panxiaole@mail.iap.ac.cn).

**Author contributions**

H.L, X.P designed the research; H.L, X.P, X.L, Y.T, Y.S, P.F, Z.W performed experiments; H.L, X.P, D.L, X.C performed the data analysis; H.L, X.P wrote the paper.

**Acknowledgements**

This study was supported by the National Natural Science Foundation of China (grant No. 41877314, 41675128).

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



**Table 1. Brief summary of some of the observations on the mixing state of rBC-containing particles.**

| rBC type | Site | Period | MMD (nm) | $D_p/D_c$ | Description | Reference |
|---|---|---|---|---|---|---|
| Urban emissions (UE) | Shenzhen, China | Aug–Sep (summer) | 180* | | The measurement station was on the university campus, located in the urban area of Shenzhen. | (Lan et al., 2013) |
| | Shenzhen, China | Jan–Feb (winter) | 210* | | The measurement station was the same as the above site. | (Huang et al., 2012) |
| | Shanghai, China | Dec (winter) | 200 | 2–8 | Maximum PM$_{2.5}$ mass loading reached 636 µg/m$^3$. | (Gong et al., 2016) |
| | Beijing, China | Feb–Mar (winter) | 213 | | The same measurement site as this study. | (Wu et al., 2017b) |
| | This study | Jun (summer) | 171 | 1.2 ($D_c$=180 nm) | | |
| | London, United Kingdom | Jan–Feb (winter) | 149 | 1.2–2 ($D_c$=110–150 nm) | During the Clean Air for London (ClearfLo) project. | (Liu et al., 2014) |
| | | Jul–Aug (summer) | 120 | | | |
| Biomass burning (BB) | Airborne measurements | Sep (autumn) | 210 | 1.33 ($D_c$=190–210 nm) | The MMD and $D_p/D_c$ were both higher for BB than UE. | (Schwarz et al., 2008) |
| | | | 189 | 1.2–1.4 ($D_c$=200 nm) | Fresh, laboratory produced biomass burning rBC. | (Pan et al., 2017) |
| | Airborne measurements | Jul–Aug (summer) | 195 | 2.35 ($D_c$=130–230 nm) | During the second phase of the BORTAS project. | (Taylor et al., 2014) |

\* Assuming the density of rBC is 2 g/cm$^3$




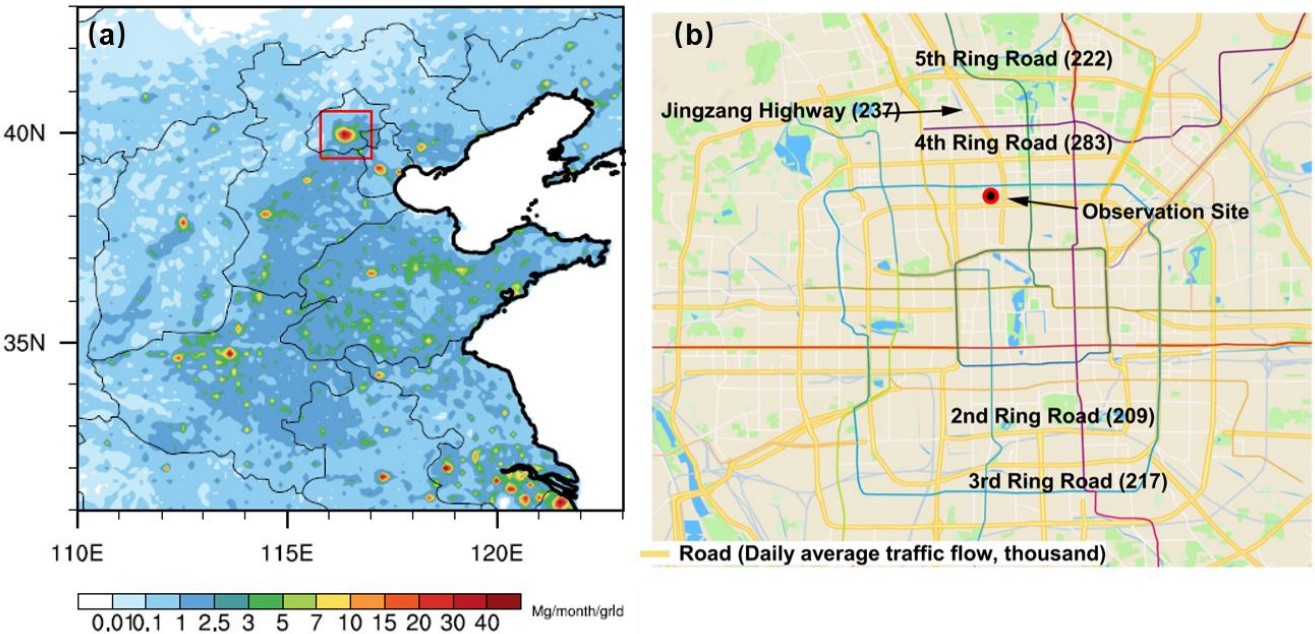

**Figure 1. (a) Monthly emissions of BC from traffic in June in east-central China. The red box denotes the geographical location of the observation site. (b) Road map and traffic flow rate of Beijing. The red circle denotes the observation site.**

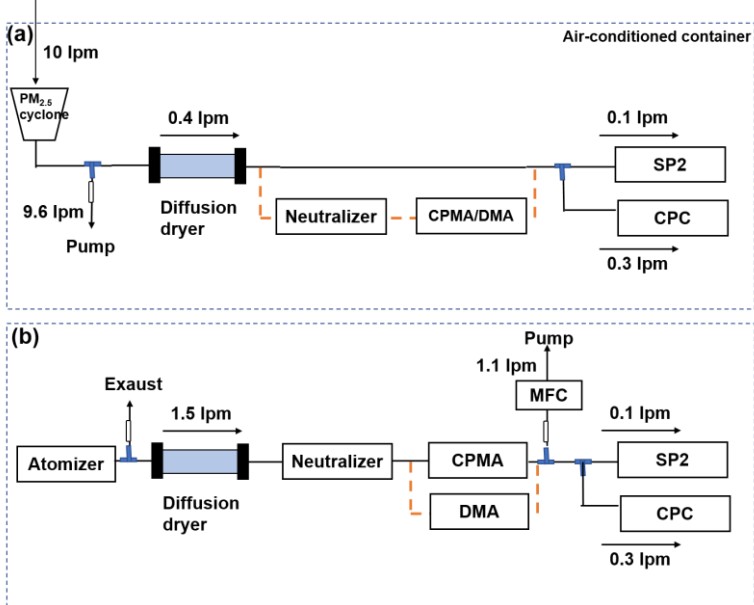

**Figure 2. Schematic diagram of (a) the measurement system, with the orange dashed line denoting the tandem CPMA/DMA-SP2 measurement system, during the periods June 8–9 and 13–14, and (b) the calibration system, with the orange dashed line denoting the calibration system using DMA.**





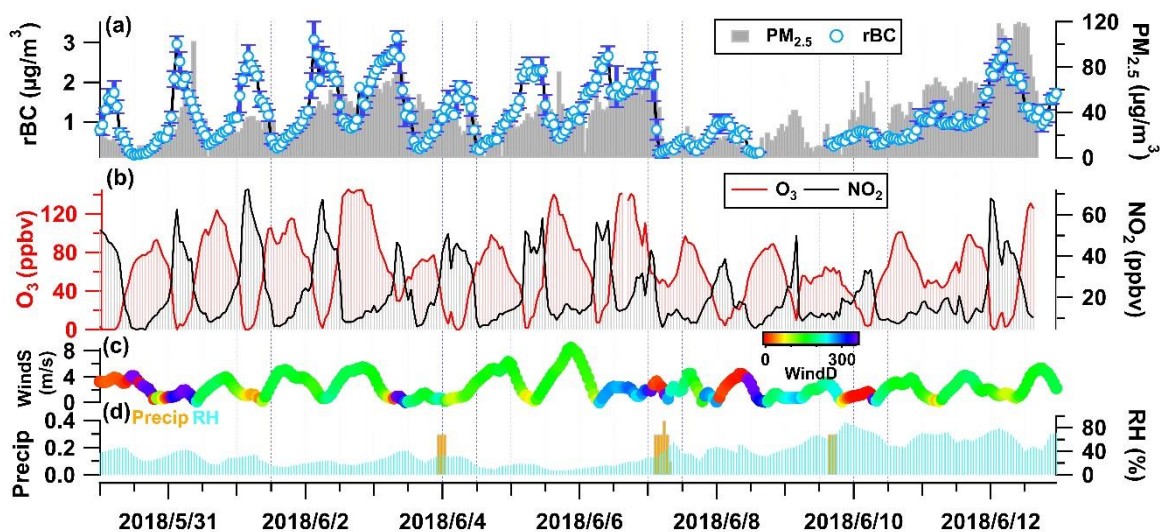

**Figure 3.** Time series of aerosol/gaseous pollutants and meteorological conditions during the observation period.

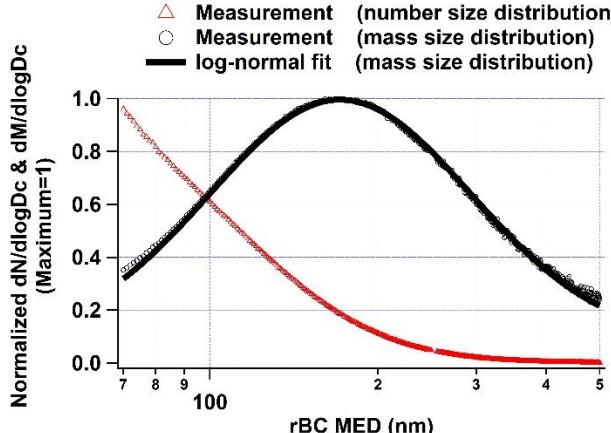

**Figure 4.** Number and mass size distribution (dN/dlog$D_c$ & dM/dlog$D_c$) during the entire project.





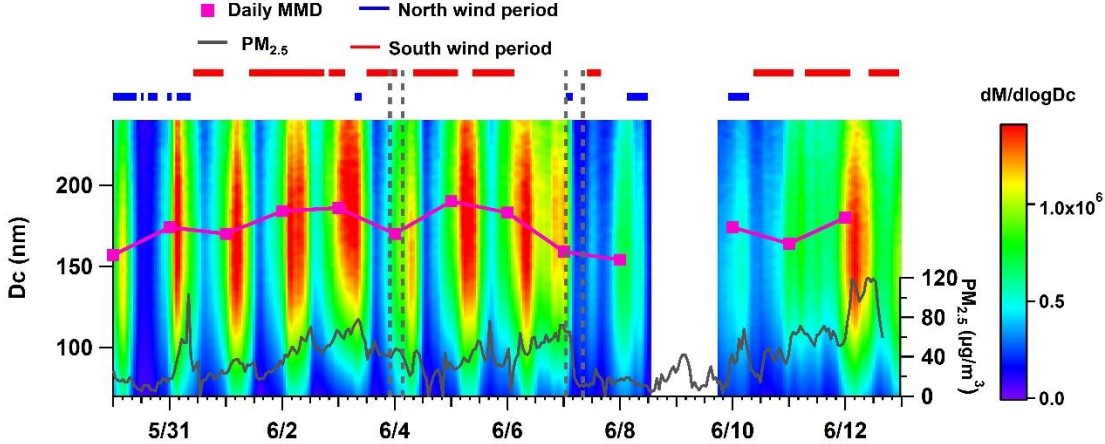

**Figure 5. Time series of the mass size distribution of rBC. The south wind period is selected when the wind direction is 135–225°**
**and the north wind period is the time when the wind direction is 325–45°. The gray dashed line denotes the rain period.**

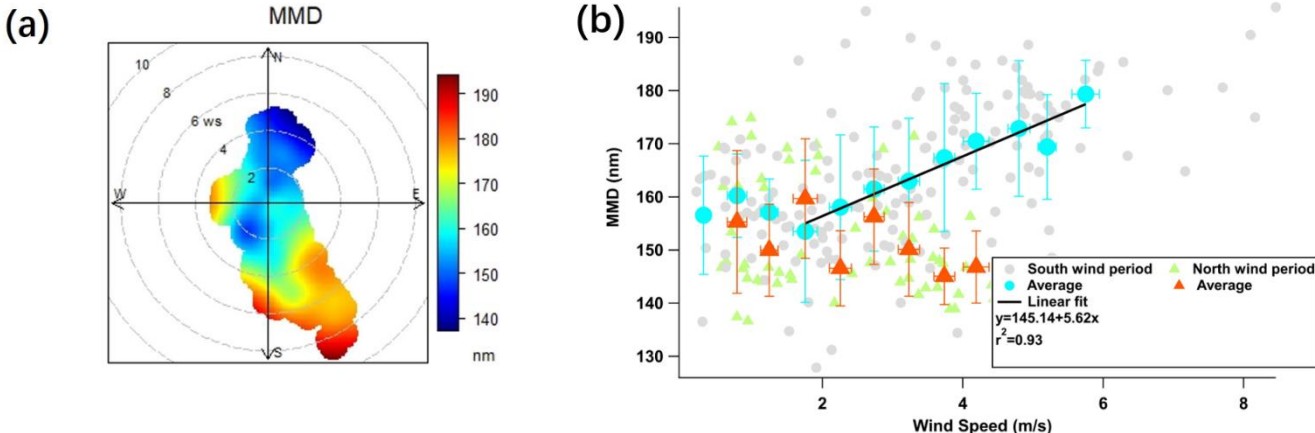

**Figure 6. (a) Dependence of rBC's MMD on wind speed and wind direction during the observation. (b) MMD versus wind speed**
**during the south wind period and north wind period. The error bars correspond to the standard deviations of MMD in each wind**
**speed bin.**





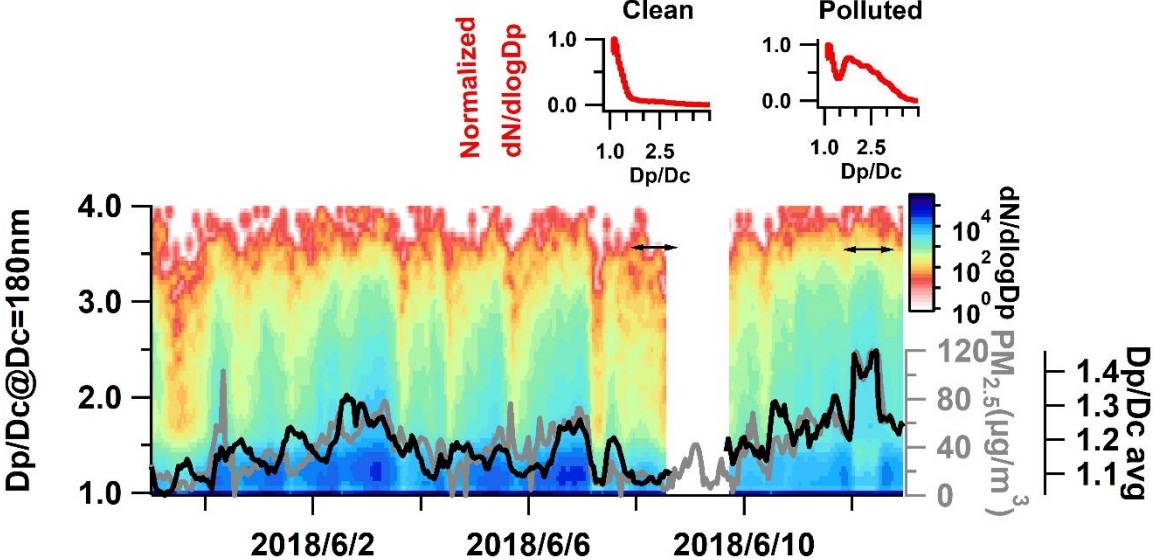

**Figure 7.** Temporal variation of $D_p/D_c$, with $D_c$=180 nm. The black line denotes the average $D_p/D_c$ for each hour. The red line on the top of the graph denotes the normalized dN/dlog$D_p$ versus $D_p/D_c$ of the clean period and pollution period.


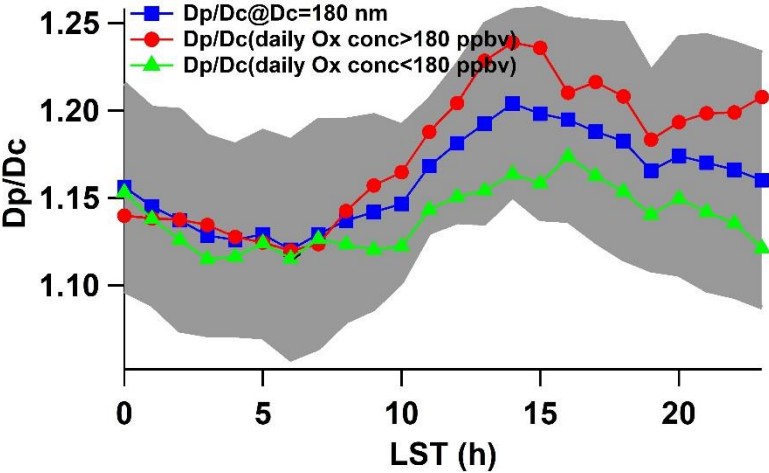

**Figure 8.** Diurnal variation of $D_p/D_c$ for all periods, Ox high period and Ox low period. The gray shaded area denotes the standard deviation of $D_p/D_c$ for all periods.



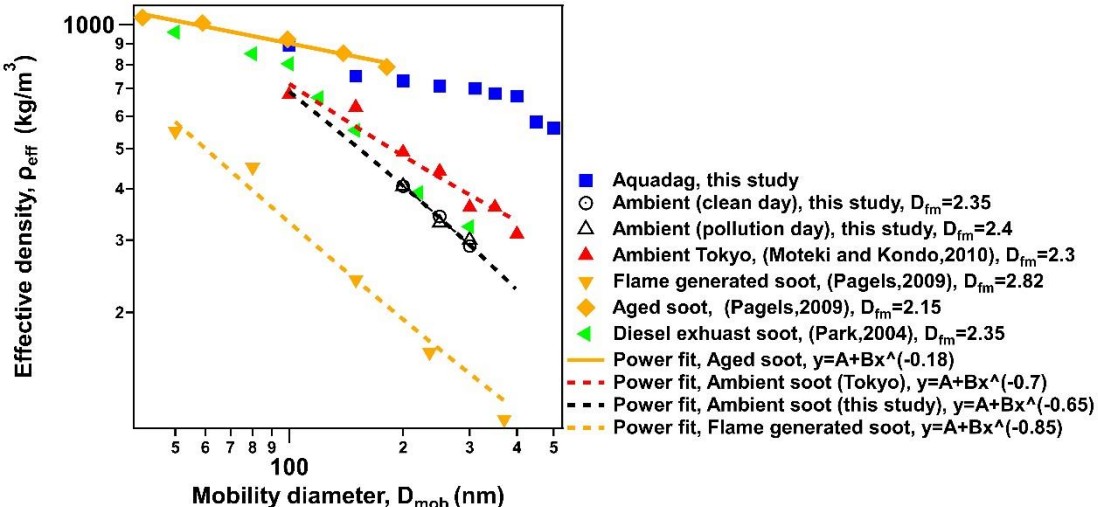


**Figure 9. Relationship between effective density and mobility diameter.**

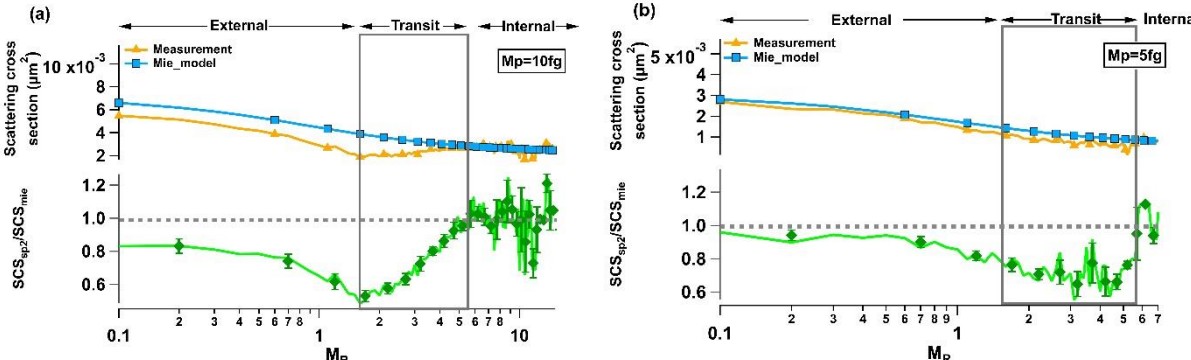


**Figure 10. Scattering cross section of rBC-containing particles measured by SP2 (yellow line) and calculated by Mie theory (blue line), assuming a core-shell structure and the ratio (green line) between these two scattering cross sections at CPMA setpoints of 5 fg and 10 fg.**



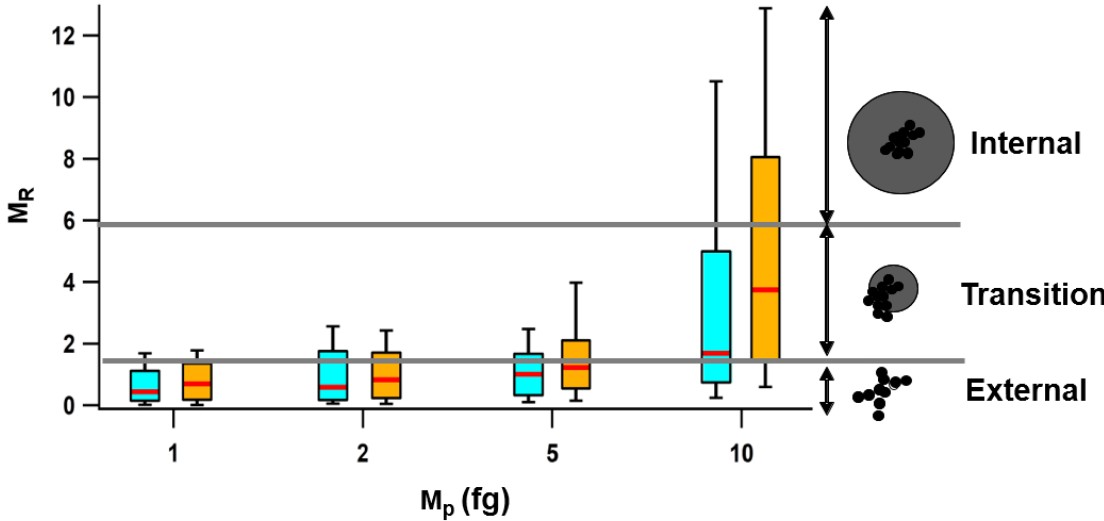

**Figure 11.** $M_R$ **under different CPMA setpoints (1 fg, 2 fg, 5 fg, 10 fg). The blue box denotes the measurement on June 8, whereas the orange box denotes the measurement on June 13 (the red lines in the middle signify the medians; the upper and lower bounds of the bars denote the 75th and 25th percentiles, respectively; and the upper and lower whiskers denote the 90th and 10th percentiles, respectively).**

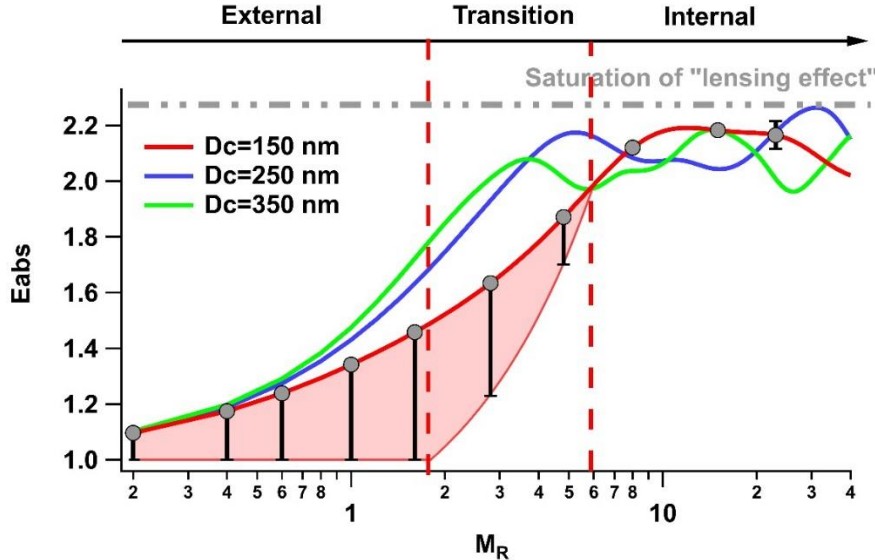


**Figure 12. Dependence of** $E_{abs}$ **on** $D_c$ **and** $M_R$ **for a wavelength at 550 nm, calculated using the Mie model. The pink shaded area denotes the uncertainty of** $E_{abs}$ **considering the morphology change.**



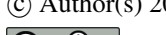

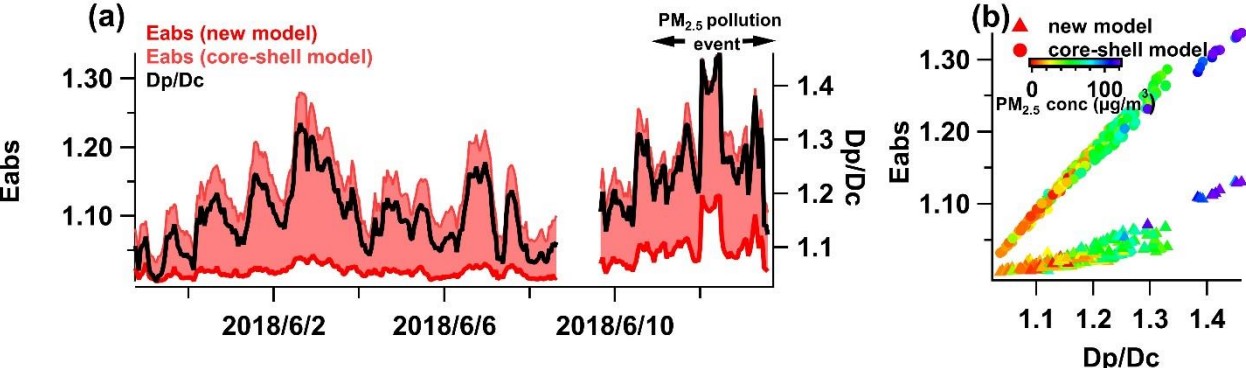

**Figure 13. (a) Time series of $D_p/D_c$ with $D_c$=180±10 nm and $E_{abs}$ at 550 nm wavelength with $D_c$=180±10 nm using the core-shell model and morphology dependent model. (b) Relationship between $E_{abs}$ and $D_p/D_c$. Circles denote the $E_{abs}$ derived from the core-shell model and triangles denote the $E_{abs}$ derived from the morphology dependent model.**