# Peer review of "Mixing characteristics of refractory black carbon aerosols determined by a tandem CPMA-SP2 system at an urban site in Beijing"

_Atmospheric Chemistry and Physics, 2019_

## Referee Comment (RC1) · Anonymous Referee #2 · 17 Jun 2019

The paper reports the microphysical properties and aging/ mixing state of rBC particles during summer time in Beijing. The research site is mostly influenced by traffic emissions from the surrounding highways and is well representative of the Beijing urban outflow. Ambient aerosol were measured using the single particle soot photometer (SP2) for ∼2 weeks (30 May to 13 June 2018). Complementary measurements of O3, NO2 and PM2.5 were performed, however, the measurement techniques were not specified in the methodology section, which I recommend to do so.

There were two case studies that the authors refers as 'clean' and 'polluted' for which the rBC properties were determined. Moreover, during these periods, a dedicated

experiment was performed by coupling a DMA or CPMA with the SP2 in order to determine the effective density, morphology and absorption enhancement of rBC particles due to coatings. The methods used in this study are valid, however, the measurement setup is questionable. For example, the authors do not mention whether the aerosol particles are dried before detection. The particle size depends on relative humidity (RH) that can strongly influence the results. Note that the RH is much higher in the "polluted" case. Moreover, I do not agree in using the terms "clean" and "polluted" applied for the two periods. The clean period is rather influenced by the fresh traffic emissions.

The description of the tandem experiment is not well described and difficult to understand. Since this is one of the highlights of the paper, it deserves a dedicated section on the methodology containing precise information of the measurement period, the atmospheric conditions during the experiment (what kind of air masses were sampled?) and the purposes of doing this. I suggest to have a dedicated section (after section 2.1) in the methodology for the case studies. A table containing the main results of this comparison can be also helpful.

Overall, I suggest improvements of the writing. In my opinion, the discussion of the results are not presented in a precise way and the figure notes are quite vague and lacking information. For all of them, I recommend to give more details, using the full name of the variables.

Specific comments:

L149: "In this study, the SP2's low detection bound was set to Dc = 70 nm". Please re-phrase.

L152: Why 1.17 factor was used?

L159: replace "owing".

L171: "The coating density was set to 1.5 g/cm3". Please re-phrase.

[Figure]

L209: remove "that".

L211-212: Which MAC value did you assume for calculating the BC mass?

L212: "Overestimation" of? Incomplete sentence.

L218: Rephrase "during which time".

L243 – 260: This whole paragraph discussing "after rain" case should be more concise. It is a bit confusing with presenting several dates. Try to group them.

L244: What was the decrease in MMD on June 4 in numbers? Is it consistent with the event on June 8?

L258: Are the southerly winds representative for the Beijing outflow? And the northerly winds?

L265: Investigation period.

L273: "Episode 1", specify the time interval.

L274: "During episode 1, the Dp/Dc distribution exhibited a single peak at 1.05. However, during episode 2, two Dp/Dc distribution peaks were found". What point do you want to make here?

L285-286: Tends vs. tended.

L304-305: There are more recent studies on the microscopy of BC.

L341: "The median MR values of the pollution day were all larger than those on the clean day for the four Mp points. This result demonstrated that rBC had more coating material during the pollution day than the clean day." Couldn't it be related to the higher RH?

L380: "indicating in-cloud nucleation scavenging may be a more efficient mechanism for rBC-containing particles". Do you mean removal mechanisms?

Fig. 1: has four panels. It is helpful for the readers if there is a full description of the measurements (from the top to the bottom) and possibly how they were collected. Moreover, I recommend to indicate the two studied cases in the figure. The same applies for figures 5.

Fig. 2: Here you present the size distribution of rBC. You applied a lognormal distribution over all the range.

Fig. 5: Y axis in log scale and increased range. What are the units of dM/dlogDc? The values seems too high!

Fig.7: In the two upper panels, the integral of the area below the curve seems to be larger than 1, is it really the normalized dN/dlogDp? Moreover, the arrows indicating the clean and polluted periods are not precise.

Fig. 8: Where are the Ox measurements from?

Fig. 9: Relationship between effective density and mobility dimeter of?

Fig. 11: MR = mass ratio of non-refractory matter to rBC. Add the full name to the figure description.

Figure S1: Does the 'after experiment' calibration have only one data point? Try to use different markers so that both measurements are visible on the plot.

Fig. S7: This figure is important to understand the origin of air masses for the two study cases. However, there is no information of the age or the starting point of the back trajectories. Also please adjust the scale of the plot.

---

## Referee Comment (RC2) · Anonymous Referee #3 · 29 Oct 2019

**General comments**

This paper reports on SP2 measurements of BC in Beijing in the summer. This dataset adds to the growing inventory of SP2 measurements and has an advantage of incorporating tandem CPMA/DMA/SP2 measurements of ambient BC.

Before final publication, there are many improvements to the writing that need to happen.

First, a full and careful proofreading is necessary to catch all the grammar and word choice errors. I have listed a bunch, but am not confident I listed them all here. In fact, after awhile, I gave up listing them because this was taking too much time. Please

correct these errors before sending out for review again.

Second, there needs to be better quantification of the SP2 calibrations. To give this dataset importance in terms of the big picture, better uncertainties are necessary. The conclusions all sound reasonable as far as I can tell; but they need to be mathematically rigorous as well.

A final big comment, there are many places in the paper that need more detail and clarifications. See my many comments listed below. In general, a better description of the tandem experiments and of the models used to calculate absorption enhancements are necessary. Many other places need rewording for clarity.

**Specific comments**

Line 21 - Is "enhancing rate" standard terminology? I don't know what the units are (0.013 what per hour?).

Line 21 - Should the "x" be a subscript in "Ox"?

Line 65 - Is the "18.97%" number really accurate to two decimal places?

Line 95 - I don't understand the sentence beginning with "Anthropogenic".

Line 99 - A few more details would be nice on Fig 2a - what is the residence time in the diffusion dryer? Did you check if there were any particle losses in the dryer? When switching configurations, how long did you wait for the sampling to stabilize?

Line 104-105 - Should show error bars/scatter in the data on Fig S1. Also should *quantify* how constant the laser was during the study. Also, be careful of wording - two data points does not ensure that the laser was constant during the study, just that the beginning and end points were similar. Without more data, it even looks like intensity may have been drifting in one direction.

Line 115 - Is it useful? (Don't use "could be".)

Line 117 - Can you estimate uncertainty from your own calibration of the SP2? You should be able to, especially with a CPMA.

Line 121 - Is Fig S2 really necessary?

Line 123 - Is it really necessary to quote the CPMA force balance equation? What does your study do with this equation specifically?

Line 127 - Is the comment about superiority of the CPMA relative to the APM necessary? What value does this statement add to your study specifically?

Line 136 - More precisely, the *peak* LII signal is what is used, not the entire LII signal.

Line 137 - Why did you use a spline fit when earlier you state that there is a linear relationship between peak height and rBC mass?

Figure S3 - Should show error bars showing the scatter in the data and uncertainty in the particle mass from the CPMA. Also, if there is no calibration equation, how do you use these data?

Line 138 - What does "approximately" mean? You should quantify these fits. And be more clear - these are spline fits like in Fig S3? My same comments apply to Fig S4 as above for S3 (include error bars, etc.).

Figure S5 - Again, would be nice to see error bars showing scattering/uncertainty in the data.

Line 152 - Why multiply by 1.17 and not 1.15?

Line 159 - This whole section should probably be edited for clarity. Specifically here, I don't understand "dividing by laster intensity".

Line 161 - Again, clarity - reword "the data before a length"

Line 173 - How did you determine which was the most proper refractive index to use?

Supplemental - What is "RCT"?

Line 182 - To be clear, the $M_{rBC}$ is what is measured by the SP2, correct? Section 2.3.3 needs some work for clarity.

Line 192 - Again, there is no quantification of how well the current study compares with previous studies. It looks like your data points are systematically higher than the polynomial fit by Gysel. You should quantify the relationship and tell the reader what it means for your study.

Line 195 - What is the purpose of Section 2.3.5? Need more details. Where exactly do the parameters going into the Mie model come from? The $C_{abs}$ variables should be defined in Table S1.

Line 202 - What instruments measured the gaseous pollutants?

Line 207 - What measured total $PM_{2.5}$ mass? Was this measurement behind the cyclone? If so, was the cyclone's cut size at 2.5 microns, or something higher? These details might effect your measurement.

Line 210 - Should provide details on the MAAP in the method section.

Line 213 - What do you mean "may be affected by coating"? With the instrumentation you have, you should be able to unambiguously determine if the coatings are the reason for the discrepancy. That analysis could be an important part of this work.

Line 213 - Maybe start a new paragraph with the sentence beginning with "During"? This paragraph is a bit haphazard and should be rewritten probably.

Line 216 - You don't specifically reference the other parts of Fig 3 - you should.

Fig S9 - Need numbers of your scale bars.

Line 223 - Why is June 13 not shown in Fig S9?

Line 223 - To this point, I still don't understand what "the tandem CPMA-DMA-SP2 experiment" is. There are a lot more details and description needed in the Methods

section.

Line 276 - It actually looks like the increase was on June 12, not June 13.

Line 277 - Where does 63% come from?

Line 295 - Do you have any idea the magnitude/emission rate of fresh rBC in Beijing? If so, you could use that number for a closure study.

Line 305 - This sentence is worded as if the Li et al 2003 study took images of the rBC from this study, which is obviously not correct. Were any new microscopy images taken from the current study period?

Line 325 - What ambient measurements? From Peng et al 2016? These effective densities are nothing like what you report in the previous section.

Line 363 - Did you find the large uncertainty? Or did Liu et al 2017? Discuss more.

Line 366 - What is the "morphology dependent model"? I am very confused by the whole Section 4.2.2.

Line 380 - More efficient than what?

**Technical corrections**

It looks like all of your citation lists are not formatted properly; missing a space.

Line 44 - should be "into the atmosphere"

Line 45 - need a comma after condensation

Line 83-84 - reword: the data are not analyzed in the discussion section, they are presented

Line 86-87 - reword "incandescent signal emissions"

Line 87 - What is "this rule"?

Line 95 - Fig 1b, specifically

Line 104 - reword: the laser intensity is not constant by performing PSL calibrations

Line 148 - unified should be unity; low should be lower

Line 151 - without should be outside

Line 156 - is intracavity a noun?

Line 164-165 - reword "description ... described"

Line 170 - the densities are not defined in the text nor in Table S1

Line 174 - add "respectively" to the sentence

Line 179 - reword

Line 205 - reword

Line 207 - Don't use "this" as the subject of a sentence.

Line 209-210 - reword, I don't think this is actually a sentence

Line 232 - MED or MMD?

Line 242 - combination should be combined

Line 243 - reword

(I've stopped marking technical corrections, though many more exist.)

---

## Author Comment (AC2) · 6 Dec 2019

**Reply to the comments of anonymous reviewer #3 on manuscript Entitled " Mixing characteristics of refractory black carbon aerosols determined by a tandem CPMA-SP2 system at an urban site in Beijing"**

We appreciate very much the patient and insight comments and recommendations of the reviewer in improving this paper and our future research. Here, we will response to all the comments one by one as follows:

General comments:

First, a full and careful proofreading is necessary to catch all the grammar and word choice errors. I have listed a bunch, but am not confident I listed them all here. In fact, after a while, I gave up listing them because this was taking too much time. Please correct these errors before sending out for review again.

Reply:Great thanks to the reviewer to point out the grammar errors. We have changed the error places and checked the manuscript several times again.

Second, there needs to be better quantification of the SP2 calibrations. To give this dataset importance in terms of the big picture, better uncertainties are necessary. The conclusions all sound reasonable as far as I can tell; but they need to be mathematically rigorous as well.

Reply: We have determined the uncertainty of our SP2 and the error bars have been added to the calibration figures. Also, the uncertainties of some key parameters have been evaluated.

A final big comment, there are many places in the paper that need more detail and clarifications. See my many comments listed below. In general, a better description of the tandem experiments and of the models used to calculate absorption enhancements are necessary. Many other places need rewording for clarity.

Reply: We have reorganized the method section and added a part to describe the tandem system including the configuration, time, the air mass during the tandem system (Section 2.32 and line 145-176 in the manuscript). The description of the calculation of absorption enhancement (Section 3 in supplementary) is added in the supplementary. Many parts of the manuscript have been rewritten to make it clearer.

Specific comments:

Line 21 - Is "enhancing rate" standard terminology? I don't know what the units are (0.013 what per hour?).

Reply: 1) We changed the expression of "enhancing rate" to "growth rate".
 2) $D_p/D_c$ is a dimensionless quantity, so there shouldn't be a unit for 0.013. We change to report the growth rate of $D_p$ instead of $D_p/D_c$ to make it clearer for readers. The growth rate of $D_p$ now has a unit of "nm". (line 21)

Line 21 - Should the "x" be a subscript in "Ox"?

Reply: Thanks, we have changed the mistake. (line 21)

Line 65 - Is the "18.97%" number really accurate to two decimal places?

Reply: We directly cited this value from the work of (Qin and Xie, 2012), so we think it's not appropriate to change the value which we cited.

Line 95 - I don't understand the sentence beginning with "Anthropogenic".

Reply: We have changed the expression to make it clearer. What we mean is the emission in the campus is little, which led little influence of the observation site. Now:

Anthropogenic emissions from the experimental campus were negligible. Thus, this site can well represent the urban conditions in Beijing. (line 89-90)

Line 99 - A few more details would be nice on Fig 2a - what is the residence time in the diffusion dryer? Did you check if there were any particle losses in the dryer? When switching configurations, how long did you wait for the sampling to stabilize?

Reply: 1) Good advice, we would like to check such loose and correct the concentration according to the probable particle loose in the diffusion dryer. Unfortunately, our SP2 has some problems now and have been sent to the manufacturers for repairing and we can't do the test at present.

2)We waited a long time (about half an hour) after changing the regular single SP2 observation to tandem system measurement. And waiting for about 2 min every time we change the setpoint of DMA/CPMA to let the system stabilize, we have added these details in the method section. (line 156, line 168).

Line 104-105 - Should show error bars/scatter in the data on Fig S1. Also should quantify how constant the laser was during the study. Also, be careful of wording – two data points does not ensure that the laser was constant during the study, just that the beginning and end points were similar. Without more data, it even looks like intensity may have been drifting in one direction.

Reply: 1) There is a parameter called YAG power which is recorded in the housekeeping file in SP2 and reflects the laser intensity. We found the YAG power was 4.8±0.1 during the observation indicating the stable condition of the laser.

2) Yes, we agree that two data points does not ensure that the laser was constant. We reword the expression and use YAG power to prove the laser was nearly constant during the study. Now:

The calibration of the scattering channel and incandescence channel was also conducted after the observation, the calibration coefficient varied little (<3%) and the YAG power (laser intensity index recorded by SP2) fluctuated with 4.8±0.1 indicating the stable condition of SP2 during the observation period. (line 119-121)

Line 115 - Is it useful? (Don't use "could be".)

Reply: We have changed the expression.

Line 117 - Can you estimate uncertainty from your own calibration of the SP2? You should be able to, especially with a CPMA.

Reply: Yes, we have determined the uncertainty of our own SP2 and reported in the manuscript. (line 105)

Line 121 - Is Fig S2 really necessary? Line 123 - Is it really necessary to quote the CPMA force balance equation? What does your study do with this equation specifically? Line 127 - Is the comment about superiority of the CPMA relative to the APM necessary? What value does this statement add to your study specifically?

Reply: We have removed the Fig S2 and simplified the introduction of CPMA to make the paper more concise.

Line 136 - More precisely, the peak LII signal is what is used, not the entire LII signal.

Reply: Yes, we have changed the expression. Now:
The LII peak-$M_{rBC}$ relationship is thus obtained (Fig. S1). (line 104)

Line 137 - Why did you use a spline fit when earlier you state that there is a linear relationship between peak height and rBC mass?

Reply: It's nearly linear relationship between LII peak height and rBC mass, but not perfectly linear. The DMT company suggested their custom to use a spline fit. The coefficient of the spline fit is exhibited in Fig S1. In fact, the coefficient of $x^2$ is very small.

Figure S3 - Should show error bars showing the scatter in the data and uncertainty in the particle mass from the CPMA. Also, if there is no calibration equation, how do you use these data?
Figure S5 - Again, would be nice to see error bars showing scattering/uncertainty in the data.

Reply: We have followed the advice of the reviewer. Now, the error bar and calibration equation are exhibited on the figure. (Fig S1-S3)

Line 138 - What does "approximately" mean? You should quantify these fits. And be more clear - these are spline fits like in Fig S3? My same comments apply to Fig S4 as above for S3 (include error bars, etc.).

Reply: The coefficient of DMA-SP2 calibration and CPMA-SP2 calibration varied little (<3%). For conciseness, we don't mention the DMA-SP2 calibration since the coefficient used in this study came from the CPMA-SP2 calibration.

Line 152 - Why multiply by 1.17 and not 1.15?

Reply: The corrected concentration is: (the measured concentration) / (1-15%) or (the measured concentration) * 1.17. We have changed the expression to "dividing by a factor of 0.85". We have changed the expression, now:
By extrapolating a lognormal function fit to the observed mass distribution, we found that rBC particles

outside the detection range caused an ~15% underestimation of the rBC mass concentration. To compensate, the mass concentration of rBC was corrected by dividing by a factor of 0.85 during the measurement. (line 128-129)

Line 159 - This whole section should probably be edited for clarity. Specifically here, I don't understand "dividing by laster intensity". Line 161 - Again, clarity - reword "the data before a length"

Reply: We have reworded the method section to make it clearer.

Line 173 - How did you determine which was the most proper refractive index to use? Supplemental - What is "RCT"?

Reply: We have removed this part and directly use the refractive index from the previous research.

Line 182 - To be clear, the $M_{rBC}$ is what is measured by the SP2, correct? Section 2.3.3 needs some work for clarity.

Reply: Yes, it's directly measured by SP2. The method section has been rewritten.

Line 192 - Again, there is no quantification of how well the current study compares with previous studies. It looks like your data points are systematically higher than the polynomial fit by Gysel. You should quantify the relationship and tell the reader what it means for your study.

Reply: Our results are ~7% higher than the poly-fit of Gysel but lower than the results from Moteki and Kondo. These differences may be result of different characteristics of Aquadag with varied lot and different instrument condition (such as the uncertainty of SP2).

Line 195 - What is the purpose of Section 2.3.5? Need more details. Where exactly do the parameters going into the Mie model come from? The Cabs variables should be defined in Table S1.

Reply: We have rewritten the method section and added a new section in the supplementary to describe the optical calculation. (Section 3 in supplementary)

Line 202 - What instruments measured the gaseous pollutants? Line 207 - What measured total PM2:5 mass? Was this measurement behind the cyclone? If so, was the cyclone's cut size at 2.5 microns, or something higher? These details might effect your measurement.

Reply: The concentrations of $PM_{2.5}$ and gaseous pollutants were from a state control air quality site, provided by the China National Environmental Monitoring Centre. The state control air quality site was 2.5 km from our observation site. We think the air quality data is similar with the air quality of our observation site in such close distance. And we added the position of the state control air quality site in Fig. 1b. (line 180-183)

Line 210 - Should provide details on the MAAP in the method section. Line 213 - What do you mean

"may be affected by coating"? With the instrumentation you have, you should be able to unambiguously determine if the coatings are the reason for the discrepancy. That analysis could be an important part of this work.

Reply: Thanks. It's a good advice to examine whether the coatings are the reason for the discrepancy. We did the test and found there wasn't strong relationship between the coating thickness and the discrepancy between MAAP and SP2. Thus, we remove the part of MAAP in the manuscript.

[Figure]

Figure 1 (a) The relationship between the mass concentration of BC (rBC) measured by MAAP and SP2, the color denoted the coating thickness. (b) The relationship between the coating thickness and the ratio of mass concentration measured by SP2 and MAAP.

Line 213 - Maybe start a new paragraph with the sentence beginning with "During"? This paragraph is a bit haphazard and should be rewritten probably.

Reply: Thanks, we changed the expression of this paragraph.

Line 216 - You don't specifically reference the other parts of Fig 3 - you should.

Reply: We try to describe more in revised version but not too much because the main purpose of this paper is to explore the properties of rBC. This Figure aimed to provide a basic observation condition for the readers.

Fig S9 - Need numbers of your scale bars. Line 223 - Why is June 13 not shown in Fig S9?

Reply: We have drawn Fig S9 again and added the backward trajectories of June 13 on Fig S9. Now: Figure S5 in supplementary.

Line 223 - To this point, I still don't understand what "the tandem CPMA-DMA-SP2 experiment" is. There are a lot more details and description needed in the Methods.

Reply: We have rewritten the method and added a new section to describe the tandem experiment. (line 145-176)

Line 276 - It actually looks like the increase was on June 12, not June 13.

Reply: Yes, it's on June 12. The episode 2 that we referred here is just June 12. We conducted the tandem experiment on June 13. So, the $D_p/D_c$ value from single SP2 measurement is only available on June 12. And we report the $D_p/D_c$ before the tandem experiment. The expression of this part has been modified. Now:

The $D_p/D_c$ distributions for the two episodes before the tandem CPMA/DMA-SP2 experiments are shown in Fig. 7. Episode 1 (June 7 2200 LST – June 8 1200 LST) occurred after a heavy rain period and is representative of clean conditions. Episode 2 (June 11 2300 LST – June 12 1200 LST) was characterized by the highest $D_p/D_c$ value (1.4) and the highest $PM_{2.5}$ concentration value (120 $\mu g/m^3$) during the observation period. (line 246-249)

Line 277 - Where does 63% come from?

Reply: 63% is cited from (Zhang et al., 2018). The air quality in Beijing is easily influenced by the regional pollution transportation in pollution conditions (Wu et al., 2017;Li et al., 2017). We just use this number in the previous research to demonstrate our inference that the rBC-containing particles with Dp/Dc = 1.8 in the right peak of the bimodal distribution may be the result of transportation from pollution region.

Line 295 - Do you have any idea the magnitude/emission rate of fresh rBC in Beijing? If so, you could use that number for a closure study.

Reply: It's a good advice. It's possible to estimate the true growth rate of $D_p$ by simultaneously considering the fresh rBC emission and the "apparent" $D_p$ growth rate. And this true growth rate value is important in the atmospheric model. Unfortunately, the emission data about the aging degree of rBC from different emission sectors is still lacking. We may conduct laboratory experiments to determine the rBC aging degree from varied rBC sources and try to estimate the true $D_p$ growth rate in the future.

Line 305 - This sentence is worded as if the Li et al 2003 study took images of the rBC from this study, which is obviously not correct. Were any new microscopy images taken from the current study period?

Reply: There was no new microscopy images taken from current study. The literature cited here is to support the argument that bare rBC is in a fractal structure. We have changed the expression to make it clearer and added some new literatures. Now:

This significant discrepancy indicates bare rBC was in a fractal structure consistent with the previous research from electron microscopic image that bare rBC was in a fractal chain-like structure (Li et al., 2003;Adachi and Buseck, 2013;Wang et al., 2017). (line 285-286)

Line 325 - What ambient measurements? From Peng et al 2016? These effective densities are nothing like what you report in the previous section.

Reply: Yes, it's from Peng et al 2016, we cite this literature in order to support the argument that the rBC-containing particles tend to become more compact with the coating increasing. The effective densities are the parameters in Peng's literature to support this argument. However, we think this sentence may confuse the readers and thus we changed the expression in the manuscript. Now:

Different techniques have been used to explore the morphology of rBC-containing particles in ambient and laboratory measurements (Zhang et al., 2008;Peng et al., 2016;Pagels et al., 2009). It is generally agreed that the morphology of rBC-containing particles will become more compact with the aging process or with increasing coating thickness. (line 310-312)

Line 363 - Did you find the large uncertainty? Or did Liu et al 2017? Discuss more.

Reply: Yes, both of our study and Liu found the uncertainty, typically in the external and transit stage because of the assumption in the morphology-dependent model. The section about the light absorption have been rephrased.

Line 366 - What is the "morphology dependent model"? I am very confused by the whole Section 4.2.2.

Reply: We rephrase this section and specifically describe the morphology-dependent model in the supplementary.

Line 380 - More efficient than what?

Reply: What we mean is the wet scavenging may be a more efficient removal mechanism for larger rBC-containing particles. We have changed the expression. Thanks for reminding.

**Technical corrections**

Great thanks for the patient and careful comments about the technical corrections from the reviewer, we have corrected the technical corrections pointed by the reviewer and carefully checked the manuscript again and again. Thanks again for the reviewer for improving this paper.

Line 44 - should be "into the atmosphere"
Line 45 - need a comma after condensation
Reply: Now: After being emitted into the atmosphere, BC particles tend to mix with other substances through coagulation, condensation, and other photochemical process, which significantly changes BC's cloud condensation nuclei activity as well as its light absorption ability (Liu et al., 2013;Bond and Bergstrom, 2006). (line 41-44)

Line 83-84 - reword: the data are not analyzed in the discussion section, they are presented
Reply: Now: A tandem experiment combining a centrifugal particle mass analyzer (CPMA, Cambustion Ltd.) and a differential mobility analyzer (DMA, model 3085A, TSI Inc., USA) with a SP2 were performed during two typical cases, focusing on BC-containing particles' microphysical properties. (line 76-78)

Line 86-87 - reword "incandescent signal emissions"
Reply: Now: After a rBC-containing particle crosses the beam, it is heated to incandesce by sequentially absorbing the laser power. (line 93-95)

Line 87 - What is "this rule"?

Reply: What we mean is we would use rBC as the abbreviation in the following section. Now:

For the SP2, the mass concentration of BC was measured on the basis of incandescent signal emissions; therefore, refractory black carbon (rBC) was used.   (line 81-82)

Line 95 - Fig 1b, specifically

Reply: Thanks, we have changed. (line 89)

Line 104 - reword: the laser intensity is not constant by performing PSL calibrations

Reply: We use a YAG power index in the housekeeping file to support the stability of our SP2. Yes, it's not constant. We rephrase the expression, now:

The calibration of the scattering channel and incandescence channel was also conducted after the observation, the calibration coefficient varied little (<3%) and the YAG power (laser intensity index recorded by SP2) fluctuated with $4.8\pm0.1$ indicating the stable condition of SP2 during the observation period. (line 119-121)

Line 148 - unified should be unity; low should be lower

Reply: Now:

For large particles, the SP2 detection efficiency was approximately unity and decreased gradually for smaller rBC particles (Fig. S3). (line 123-124)

Line 151 - without should be outside

Reply: Now:

By extrapolating a lognormal function fit to the observed mass distribution, we found that rBC particles outside the detection range caused an ~15% underestimation of the rBC mass concentration. (line 126-128)

Line 156 - is intracavity a noun?

Line 164-165 - reword "description ... described"

Line 170 - the densities are not defined in the text nor in Table S1

Line 174 - add "respectively" to the sentence

Line 179 - reword

Reply: These sentences have been deleted in the method section in the new manuscript.

Line 205 – reword

Reply: Former: $O_3$ dominant pollution occurred at 1400 LST on June 2, with a maximum of 145 ppbv, reflecting high atmospheric oxidant levels and strong photochemistry during the observation.

Now: The maximum $O_3$ concentration appeared at 1400 LST on June 2 with a value of 145 ppbv, reflecting high atmospheric oxidant levels and strong photochemistry during the observation. (line 185)

Line 207 - Don't use "this" as the subject of a sentence.

Reply: Former:" The mass concentration of rBC was $1.21 \pm 0.73$ $\mu g/m^3$ on average, accounting for $3.5 \pm 2.4\%$ of $PM_{2.5}$ on an hourly basis. This was comparable to the previous filter-based measurement in Beijing, with an average fraction of 3.2% in the summer of 2010 (Zhang et al., 2013)."

Now: The mass concentration of rBC was $1.21 \pm 0.73$ μg/m³ on average, accounting for $3.5 \pm 2.4\%$ of $PM_{2.5}$ on an hourly basis, which was comparable to the previous filter-based measurement in Beijing, with an average fraction of 3.2% in the summer of 2010 (Zhang et al., 2013). (line 186-188)

Line 209-210 - reword, I don't think this is actually a sentence
Reply: It has been deleted.

Line 232 - MED or MMD?
Reply: It's MED. Liu et al. (2014) only reported the average MED and standard deviation of MED in winter and summer but not MMD, but these MED value also show a summer-low-winter-high trend.

Line 242 - combination should be combined
Reply: Now:
The diurnal cycle reached a peak plateau between 0300–0700 LST and it decreased gradually in the afternoon, which was controlled by the combined effects of the development of a planetary boundary layer (PBL) variation and on-road rBC emissions. (line 213-215)

Line 243 – reword
Reply: Former: "Significant change of rBC's mass size distribution occurred on June 7, corresponding to the heavy rain period. After the heavy rain event, the MMD decreased sharply to 159 nm. This decreasing trend of MMD also occurred on June 4 during another rainfall event."
Now: After the two rain events (June 4 and June 7) as shown in Fig. 3, the MMD decreased significantly from 186 nm to 170 nm and from 183 nm to 159 nm separately. (line 216-217)

Li, J., Du, H., Wang, Z., Sun, Y., Yang, W., Li, J., Tang, X., and Fu, P.: Rapid formation of a severe regional winter haze episode over a mega-city cluster on the North China Plain, Environ Pollut, 223, 605-615, 2017.

Liu, D., Allan, J. D., Young, D. E., Coe, H., Beddows, D., Fleming, Z. L., Flynn, M. J., Gallagher, M. W., Harrison, R. M., Lee, J., Prevot, A. S. H., Taylor, J. W., Yin, J., Williams, P. I., and Zotter, P.: Size distribution, mixing state and source apportionment of black carbon aerosol in London during wintertime, Atmospheric Chemistry and Physics, 14, 10061-10084, 10.5194/acp-14-10061-2014, 2014.

Pagels, J., Khalizov, A. F., McMurry, P. H., and Zhang, R. Y.: Processing of Soot by Controlled Sulphuric Acid and Water CondensationMass and Mobility Relationship, Aerosol Science and Technology, 43, 629-640, Pii 910341827
10.1080/02786820902810685, 2009.

Peng, J. F., Hu, M., Guo, S., Du, Z. F., Zheng, J., Shang, D. J., Zamora, M. L., Zeng, L. M., Shao, M., Wu, Y. S., Zheng, J., Wang, Y., Glen, C. R., Collins, D. R., Molina, M. J., and Zhang, R. Y.: Markedly enhanced absorption and direct radiative forcing of black carbon under polluted urban environments, P Natl Acad Sci USA, 113, 4266-4271, 10.1073/pnas.1602310113, 2016.

Qin, Y., and Xie, S. D.: Spatial and temporal variation of anthropogenic black carbon emissions in China for the period 1980-2009, Atmospheric Chemistry and Physics, 12, 4825-4841, 10.5194/acp-12-4825-2012, 2012.

Wu, J. B., Wang, Z. F., Wang, Q., Li, J., Xu, J. M., Chen, H. S., Ge, B. Z., Zhou, G. Q., and Chang, L. Y.: Development of an on-line source-tagged model for sulfate, nitrate and ammonium: A modeling study for highly polluted periods in Shanghai, China, Environ Pollut, 221, 168-179, 10.1016/j.envpol.2016.11.061, 2017.

Zhang, R. Y., Khalizov, A. F., Pagels, J., Zhang, D., Xue, H. X., and McMurry, P. H.: Variability in morphology, hygroscopicity, and optical properties of soot aerosols during atmospheric processing, P Natl Acad Sci USA, 105, 10291-10296, 10.1073/pnas.0804860105, 2008.

Zhang, Y. X., Zhang, Q., Cheng, Y. F., Su, H., Li, H. Y., Li, M., Zhang, X., Ding, A. J., and He, K. B.: Amplification of light absorption of black carbon associated with air pollution, Atmospheric Chemistry and Physics, 18, 9879-9896, 10.5194/acp-18-9879-2018, 2018.

---

## Author Response (AR1)

**Reply to the comments of anonymous reviewer #2 on manuscript Entitled " Mixing characteristics of refractory black carbon aerosols determined by a tandem CPMA-SP2 system at an urban site in Beijing"**

We appreciate very much the patient and insight comments and recommendations of the reviewer in improving this paper and our future research. Here, we will response to all the comments one by one as follows:

General comments:

The paper reports the microphysical properties and aging/ mixing state of rBC particles during summer time in Beijing. The research site is mostly influenced by traffic emissions from the surrounding highways and is well representative of the Beijing urban outflow. Ambient aerosol were measured using the single particle soot photometer (SP2) for 2 weeks (30 May to 13 June 2018). Complementary measurements of $O_3$, $NO_2$ and $PM_{2.5}$ were performed, however, the measurement techniques were not specified in the methodology section, which I recommend to do so.

Reply: The concentrations of $PM_{2.5}$ and gaseous pollutants were from a state control air quality site, provided by the China National Environmental Monitoring Centre. The state control air quality site was 2.5 km from our observation site. We think the air quality condition of the state control site was similar with the air quality condition of our observation site in such close distance. And we added the position of the state control air quality site in Fig. 1b and the measuring instrument type for every pollutants (line 180-183).

There were two case studies that the authors refers as 'clean' and 'polluted' for which the rBC properties were determined. Moreover, during these periods, a dedicated experiment was performed by coupling a DMA or CPMA with the SP2 in order to determine the effective density, morphology and absorption enhancement of rBC particles due to coatings. The methods used in this study are valid, however, the measurement setup is questionable. For example, the authors do not mention whether the aerosol particles are dried before detection. The particle size depends on relative humidity (RH) that can strongly influence the results. Note that the RH is much higher in the "polluted" case.

Reply: We add a specific description about the tandem system in a new section to make it more clear, line (145-176). The air sample was dried before it entered into the tandem system. Thus, the RH wouldn't influence the particle size in this study (Fig. 2a).

Moreover, I do not agree in using the terms "clean" and "polluted" applied for the two periods. The clean period is rather influenced by the fresh traffic emissions.

Reply: The clean and polluted periods were defined according to the Chinese air quality standard which mainly focus on the mass concentration of $PM_{2.5}$. We agree there is still substantial rBC emission in clean periods. However, there were more

pollutant gases and particle matters in polluted periods which may influence the rBC-containing particles' coating and morphology. For example, Zhang et al. (2018) found the $D_p/D_c$ value of rBC-containing particles tended to be larger in the episodes with higher $PM_{2.5}$ mass concentration. That's Why we conducted the tandem experiments separately in clean and polluted episodes.

The description of the tandem experiment is not well described and difficult to understand. Since this is one of the highlights of the paper, it deserves a dedicated section on the methodology containing precise information of the measurement period, the atmospheric conditions during the experiment (what kind of air masses were sampled?) and the purposes of doing this. I suggest to have a dedicated section (after section 2.1) in the methodology for the case studies. A table containing the main results of this comparison can be also helpful.

Reply: Thanks for the advice, we added a new section to describe the tandem system, (line 145-176).

Overall, I suggest improvements of the writing. In my opinion, the discussion of the results are not presented in a precise way and the figure notes are quite vague and lacking information. For all of them, I recommend to give more details, using the full name of the variables.

Reply: Thanks for the comments, we will improve our writing in the following version. The figure notes and legends have been improved.

Specific comments:

L149: "In this study, the SP2's low detection bound was set to Dc = 70 nm". Please re-phrase.

Reply: The sentence has been changed to

The former : "In this study, the SP2's low detection bound was set to $D_c = 70$ nm." (line 149)

Now: "For rBC with Dc< 70 nm, the detect efficiency of SP2 significantly dropped below 60%. Thus, rBC with Dc< 70 nm was not considered in this study." (line 124-126)

L152: Why 1.17 factor was used?

Reply: The corrected concentration is : (the measured concentration) / (1-15%) which is the same as (the measured concentration) * 1.17. We have changed the expression:

By extrapolating a lognormal function fit to the observed mass distribution, we found that rBC particles outside the detection range caused an ~15% underestimation of the rBC mass concentration. To compensate, the mass concentration of rBC was corrected by dividing by a factor of 0.85 during the measurement. (line 126-129)

L159: replace "owing".

Reply: The sentence has been rewritten.

The former: "For rBC-containing particles, the scattering cross section significantly decreased owing to the evaporation of the nonrefractory coating as the rBC core absorbed energy" (line 159)

Now: "However, for rBC-containing particles, the particles will evaporate during the measurement since rBC can absorb the laser energy, which results in a decrease in the rBC-containing particles' sizes and thus a decrease in the $\sigma_{measured}$." (line 108-110)

L171: "The coating density was set to 1.5 g/cm3". Please re-phrase.

Reply: The sentence has been rewritten.

The former: "The coating density was set to 1.5 g/cm³". (line 171)

Now: "Knowing $M_P$ and $M_{rBC}$, the scattering cross section of rBC-containing particles can be calculated through Mie-theory with refractive indices of 2.26-1.26i of rBC and 1.48i of coatings, by assuming a core-shell structure and the coating density of 1.5 g/cm$^3$." (line 172-174)

L209: remove "that".

Reply: We have removed "that" in the text.

L211-212: Which MAC value did you assume for calculating the BC mass? L212: "Overestimation" of? Incomplete sentence.

Reply: The discussion about MAAP have been removed in the new manuscript since we found the coatings weren't the cause of the bias of the rBC mass concentration measured by MAAP and SP2

L218: Rephrase "during which time".

Reply: The sentence have been rewritten.

The former: "A heavy rainfall event occurred from 0300‒0700LST on June 7, during which time the mass concentration of PM$_{2.5}$ decreased from 65 to 10 μg/m$_3$ and the mass concentration of rBC decreased from 2.63 to 0.2 μg/m$^3$. (line 218)

Now: " On June 7, a heavy rain fall event occurred, most of the major pollutants decreased due to significant wet scavenging. The mass concentration of PM$_{2.5}$ decreased from 65 to 10 μg/m3 and the mass concentration of rBC decreased from 2.63 to 0.2 μg/m$^3$ from 0300–0700 LST on June 7." (line 190-192)

L243 – 260: This whole paragraph discussing "after rain" case should be more concise. It is a bit confusing with presenting several dates. Try to group them.

Reply: Thanks for the advice, we have rewritten the paragraph and make the paragraph more concise. (line 216-221)

 What was the decrease in MMD on June 4 in numbers? Is it consistent with the event on June 8?

Reply: The MMD decreased from 186 nm to 170 nm on June 4 and decreased from 183 nm to 159 nm on June 8 (Figure 5). The more decrease of MMD on June 8 may be the result of heavier rain event (Figure 3). We have added the specific MMD number in the paragraph. (line 216)

L258: Are the southerly winds representative for the Beijing outflow? And the northerly winds?

Reply: Beijing outflow is mainly affected by the southerly wind and the northerly winds. The north of Beijing is a clean region with little emission while the south of Beijing is one of the most polluted regions in China. We add a detailed discussion about the MMD characteristic when Beijing was affected by the northerly winds to make the result more comprehensive. Since the air mass from the north is always clean, the local rBC emissions may be the main contributors to the total rBC concentration in the northerly wind period. Thus, the MMD may be more influenced by local emissions and show a weak correlation with the wind speed during northerly wind periods. (line 230-234)

L265: Investigation period.

Reply: We have changed the expression

L273: "Episode 1", specify the time interval.

Reply: We have specified the time interval in the text. The Dp/Dc distributions for the two episodes before the tandem CPMA/DMA-SP2 experiments are shown in Fig. 7. Episode 1 (June 7 2200 LST – June 8 1200 LST) occurred after a heavy rain period and is representative of a clean condition. Episode 2 (June 11 2300 LST – June 12 1200 LST) was characterized with the highest Dp/Dc value (1.4) and the highest $PM_{2.5}$ concentration value (120 $\mu g/m^3$). (line 246-249)

L274: "During episode 1, the Dp/Dc distribution exhibited a single peak at 1.05. However, during episode 2, two Dp/Dc distribution peaks were found". What point do you want to make here?

Reply: We want to exhibit the different $D_p/D_c$ distribution during episode 1 and episode 2. We have changed the expression which may be more understandable and concise. The Dp/Dc exhibited a unimodal distribution during episode 1 and a clear bimodal pattern during episode 2 as shown in the upper panel of Fig. 7. (line 249-250)

L285-286: Tends vs. tended.

Reply: Thanks, "tends" seems to be more appropriate.

L304-305: There are more recent studies on the microscopy of BC.

135      Reply: It's true. We update the reference now. (line 282)

L341: "The median MR values of the pollution day were all larger than those on the clean day for the four Mp points. This result demonstrated that rBC had more coating material during the pollution day than the clean day." Couldn't it be related to the higher RH?

140      Reply: A higher RH can truly increase the water content in the rBC-containing particles and thus the size as well as $M_R$. However, we have dried the rBC-containing particles before the tandem system as shown in Fig. 2(a) to avoid the influence of RH. We will write a new section to introduce the tandem system including the drying process to avoid causing misunderstanding of readers.

145
Reply: Yes, we will add "removal" to make it more clear.

150
Reply: A new section which is used to describe the tandem system has been added. (line 145-176) and the two studied cases have been indicated in the figure notes of Fig. 2. (line 546-547)
The Fig. 3 has been improved to denote the time when the tandem system was conducted.

155

Reply: Thanks for the advice, we have changed the figure. The unit of $dM/dlogD_c$ has been changed to µg m$^{-3}$.

160
Reply: The former normalized dN/dlogDp was obtained by letting the maximum value of the histogram be 1. We have changed the calculation to let the integral of the area be 1.
The arrows have been in bold to make them more clear and extra guide line have been added to denote the clean and polluted periods.

165

Reply: The concentrations of $PM_{2.5}$ and gaseous pollutants were from a state control air quality site, provided by the China National Environmental Monitoring Centre. The state control air quality site was 2.5 km from our observation site. We think

the air quality data is similar in such close distance. And we added the position of the state control air quality site in Fig. 1b.
(line 186-189)

Fig. 9: Relationship between effective density and mobility dimeter of?

Reply: We have made the figure note more specific. Now:

==Relationship between effective density and mobility diameter of rBC-containing particles. The black circle and triangle denote the fresh rBC-containing particles (Dp/Dc = 1) measured in clean day and polluted day in this study. Other markers denote the data from previous research.== (line 576-579)

Fig. 11: MR = mass ratio of non-refractory matter to rBC. Add the full name to the figure description.

Reply: We have followed your advice. (line 584)

Figure S1: Does the 'after experiment' calibration have only one data point? Try to use different markers so that both measurements are visible on the plot.

Reply: Yes, there is only one data point. The purpose of this calibration is to determine the laser intensity of SP2. In fact, the DMT company suggests the calibration of the scattering signal only needs one data point. Thus, we only did calibration of scattering signal using PSL with diameter of 240 nm after the experiment which showed good consistency with the calibration before the experiment (varied within 3%) demonstrating the stability of the instrument.

Fig. S7: This figure is important to understand the origin of air masses for the two study cases. However, there is no information of the age or the starting point of the back trajectories. Also please adjust the scale of the plot.

Reply: We have drawn this figure again and specifically describe the calculation of the trajectories. (Section 2 in Supplementary, line 45-54).

Zhang, Y. X., Zhang, Q., Cheng, Y. F., Su, H., Li, H. Y., Li, M., Zhang, X., Ding, A. J., and He, K. B.: Amplification of light absorption of black carbon associated with air pollution, Atmospheric Chemistry and Physics, 18, 9879-9896, 10.5194/acp-18-9879-2018, 2018.

**Reply to the comments of anonymous reviewer #3 on manuscript Entitled " Mixing characteristics of refractory black carbon aerosols determined by a tandem CPMA-SP2 system at an urban site in Beijing"**

We appreciate very much the patient and insight comments and recommendations of the reviewer in improving this paper and our future research. Here, we will response to all the comments one by one as follows:

General comments:

First, a full and careful proofreading is necessary to catch all the grammar and word choice errors. I have listed a bunch, but am not confident I listed them all here. In fact, after a while, I gave up listing them because this was taking too much time. Please correct these errors before sending out for review again.

Reply:Great thanks to the reviewer to point out the grammar errors. We have changed the error places and checked the manuscript several times again.

Second, there needs to be better quantification of the SP2 calibrations. To give this dataset importance in terms of the big picture, better uncertainties are necessary. The conclusions all sound reasonable as far as I can tell; but they need to be mathematically rigorous as well.

Reply: We have determined the uncertainty of our SP2 and the error bars have been added to the calibration figures. Also, the uncertainties of some key parameters have been evaluated.

A final big comment, there are many places in the paper that need more detail and clarifications. See my many comments listed below. In general, a better description of the tandem experiments and of the models used to calculate absorption enhancements are necessary. Many other places need rewording for clarity.

Reply: We have reorganized the method section and added a part to describe the tandem system including the configuration, time, the air mass during the tandem system (Section 2.32 and line 145-176 in the manuscript). The description of the calculation of absorption enhancement (Section 3 in supplementary) is added in the supplementary. Many parts of the manuscript have been rewritten to make it clearer.

Specific comments:

Line 21 - Is "enhancing rate" standard terminology? I don't know what the units are (0.013 what per hour?).

Reply: 1) We changed the expression of "enhancing rate" to "growth rate".

2) $D_p/D_c$ is a dimensionless quantity, so there shouldn't be a unit for 0.013. We change to report the growth rate of $D_p$ instead of $D_p/D_c$ to make it clearer for readers. The growth rate of $D_p$ now has a unit of "nm". (line 21)

Line 21 - Should the "x" be a subscript in "Ox"?

Reply: Thanks, we have changed the mistake. (line 21)

Line 65 - Is the "18.97%" number really accurate to two decimal places?

Reply: We directly cited this value from the work of (Qin and Xie, 2012), so we think it's not appropriate to change the value which we cited.

Line 95 - I don't understand the sentence beginning with "Anthropogenic".

Reply: We have changed the expression to make it clearer. What we mean is the emission in the campus is little, which led little influence of the observation site. Now:

==Anthropogenic emissions from the experimental campus were negligible. Thus, this site can well represent the urban conditions in Beijing. (line 89-90)==

Line 99 - A few more details would be nice on Fig 2a - what is the residence time in the diffusion dryer? Did you check if there were any particle losses in the dryer? When switching configurations, how long did you wait for the sampling to stabilize?

Reply: 1) Good advice, we would like to check such loose and correct the concentration according to the probable particle loose in the diffusion dryer. Unfortunately, our SP2 has some problems now and have been sent to the manufacturers for repairing and we can't do the test at present.

2)We waited a long time (about half an hour) after changing the regular single SP2 observation to tandem system measurement. And waiting for about 2 min every time we change the setpoint of DMA/CPMA to let the system stabilize, we have added these details in the method section. (line 156, line 168).

Line 104-105 - Should show error bars/scatter in the data on Fig S1. Also should quantify how constant the laser was during the study. Also, be careful of wording – two data points does not ensure that the laser was constant during the study, just that the beginning and end points were similar. Without more data, it even looks like intensity may have been drifting in one direction.

Reply: 1) There is a parameter called YAG power which is recorded in the housekeeping file in SP2 and reflects the laser intensity. We found the YAG power was $4.8\pm0.1$ during the observation indicating the stable condition of the laser.

265   2) Yes, we agree that two data points does not ensure that the laser was constant. We reword the expression and use YAG power to prove the laser was nearly constant during the study. Now:

==The calibration of the scattering channel and incandescence channel was also conducted after the observation, the calibration coefficient varied little (<3%) and the YAG power (laser intensity index recorded by SP2) fluctuated with 4.8±0.1 indicating the stable condition of SP2 during the observation period.== (line 119-121)

270

Line 115 - Is it useful? (Don't use "could be".)

Reply: We have changed the expression.

275   Line 117 - Can you estimate uncertainty from your own calibration of the SP2? You should be able to, especially with a CPMA.

Reply: Yes, we have determined the uncertainty of our own SP2 and reported in the manuscript. (line 105)

280   Line 121 - Is Fig S2 really necessary? Line 123 - Is it really necessary to quote the CPMA force balance equation? What does your study do with this equation specifically? Line 127 - Is the comment about superiority of the CPMA relative to the APM necessary? What value does this statement add to your study specifically?

Reply: We have removed the Fig S2 and simplified the introduction of CPMA to make the paper more concise.

285   Line 136 - More precisely, the peak LII signal is what is used, not the entire LII signal.

Reply: Yes, we have changed the expression. Now:

==The LII peak-$M_{rBC}$ relationship is thus obtained (Fig. S1). (line 104)==

290   Line 137 - Why did you use a spline fit when earlier you state that there is a linear relationship between peak height and rBC mass?

Reply: It's nearly linear relationship between LII peak height and rBC mass, but not perfectly linear. The DMT company suggested their custom to use a spline fit. The coefficient of the spline fit is exhibited in Fig S1. In fact, the coefficient of $x^2$ is

295   very small.

Figure S3 - Should show error bars showing the scatter in the data and uncertainty in the particle mass from the CPMA. Also, if there is no calibration equation, how do you use these data?

300

Figure S5 - Again, would be nice to see error bars showing scattering/uncertainty in the data.

Reply: We have followed the advice of the reviewer. Now, the error bar and calibration equation are exhibited on the figure. (Fig S1-S3)

305

Line 138 - What does "approximately" mean? You should quantify these fits. And be more clear - these are spline fits like in Fig S3? My same comments apply to Fig S4 as above for S3 (include error bars, etc.).

Reply: The coefficient of DMA-SP2 calibration and CPMA-SP2 calibration varied little (<3%). For conciseness, we don't mention the DMA-SP2 calibration since the coefficient used in this study came from the CPMA-SP2 calibration.

310

Reply: The corrected concentration is: (the measured concentration) / (1-15%) or (the measured concentration) * 1.17. We have changed the expression to "dividing by a factor of 0.85". We have changed the expression, now:
By extrapolating a lognormal function fit to the observed mass distribution, we found that rBC particles outside the detection

315 range caused an ~15% underestimation of the rBC mass concentration. To compensate, the mass concentration of rBC was corrected by dividing by a factor of 0.85 during the measurement. (line 128-129)

320

Line 159 - This whole section should probably be edited for clarity. Specifically here, I don't understand "dividing by laster intensity". Line 161 - Again, clarity - reword "the data before a length"

Reply: We have reworded the method section to make it clearer.

Line 173 - How did you determine which was the most proper refractive index to use? Supplemental - What is "RCT"?

325 Reply: We have removed this part and directly use the refractive index from the previous research.

Line 182 - To be clear, the $M_{rBC}$ is what is measured by the SP2, correct? Section 2.3.3 needs some work for clarity.

Reply: Yes, it's directly measured by SP2. The method section has been rewritten.

330

Line 192 - Again, there is no quantification of how well the current study compares with previous studies. It looks like your data points are systematically higher than the polynomial fit by Gysel. You should quantify the relationship and tell the reader what it means for your study.

335    Reply: Our results are ~7% higher than the poly-fit of Gysel but lower than the results from Moteki and Kondo. These differences may be result of different characteristics of Aquadag with varied lot and different instrument condition (such as the uncertainty of SP2).

Line 195 - What is the purpose of Section 2.3.5? Need more details. Where exactly do the parameters going into the Mie model
340    come from? The Cabs variables should be defined in Table S1.

Reply: We have rewritten the method section and added a new section in the supplementary to describe the optical calculation. (Section 3 in supplementary)

345    Line 202 - What instruments measured the gaseous pollutants? Line 207 - What measured total PM2:5 mass? Was this measurement behind the cyclone? If so, was the cyclone's cut size at 2.5 microns, or something higher? These details might effect your measurement.

Reply: The concentrations of $PM_{2.5}$ and gaseous pollutants were from a state control air quality site, provided by the China
350    National Environmental Monitoring Centre. The state control air quality site was 2.5 km from our observation site. We think the air quality data is similar with the air quality of our observation site in such close distance. And we added the position of the state control air quality site in Fig. 1b. (line 180-183)

Line 210 - Should provide details on the MAAP in the method section. Line 213 - What do you mean "may be affected by
355    coating"? With the instrumentation you have, you should be able to unambiguously determine if the coatings are the reason for the discrepancy. That analysis could be an important part of this work.

Reply: Thanks. It's a good advice to examine whether the coatings are the reason for the discrepancy. We did the test and found there wasn't strong relationship between the coating thickness and the discrepancy between MAAP and SP2. Thus, we
360    remove the part of MAAP in the manuscript.

[Figure]

Figure 1 (a) The relationship between the mass concentration of BC (rBC) measured by MAAP and SP2, the color denoted the coating thickness. (b) The relationship between the coating thickness and the ratio of mass concentration measured by SP2 and MAAP.

Line 213 - Maybe start a new paragraph with the sentence beginning with "During"? This paragraph is a bit haphazard and should be rewritten probably.

Reply: Thanks, we changed the expression of this paragraph.

Line 216 - You don't specifically reference the other parts of Fig 3 - you should.

Reply: We try to describe more in revised version but not too much because the main purpose of this paper is to explore the properties of rBC. This Figure aimed to provide a basic observation condition for the readers.

Fig S9 - Need numbers of your scale bars. Line 223 - Why is June 13 not shown in Fig S9?

Reply: We have drawn Fig S9 again and added the backward trajectories of June 13 on Fig S9. Now: Figure S5 in supplementary.

Line 223 - To this point, I still don't understand what "the tandem CPMA-DMA-SP2 experiment" is. There are a lot more details and description needed in the Methods.

Reply: We have rewritten the method and added a new section to describe the tandem experiment. (line 145-176)

Line 276 - It actually looks like the increase was on June 12, not June 13.

Reply: Yes, it's on June 12. The episode 2 that we referred here is just June 12. We conducted the tandem experiment on June 13. So, the $D_p/D_c$ value from single SP2 measurement is only available on June 12. And we report the $D_p/D_c$ before the tandem experiment. The expression of this part has been modified. Now:

The $D_p/D_c$ distributions for the two episodes before the tandem CPMA/DMA-SP2 experiments are shown in Fig. 7. Episode 1 (June 7 2200 LST – June 8 1200 LST) occurred after a heavy rain period and is representative of clean conditions. Episode 2 (June 11 2300 LST – June 12 1200 LST) was characterized by the highest $D_p/D_c$ value (1.4) and the highest $PM_{2.5}$ concentration value (120 µg/m$^3$) during the observation period. (line 246-249)

Line 277 - Where does 63% come from?

Reply: 63% is cited from (Zhang et al., 2018). The air quality in Beijing is easily influenced by the regional pollution transportation in pollution conditions (Wu et al., 2017;Li et al., 2017). We just use this number in the previous research to demonstrate our inference that the rBC-containing particles with $D_p/D_c$ = 1.8 in the right peak of the bimodal distribution may be the result of transportation from pollution region.

Line 295 - Do you have any idea the magnitude/emission rate of fresh rBC in Beijing? If so, you could use that number for a closure study.

Reply: It's a good advice. It's possible to estimate the true growth rate of $D_p$ by simultaneously considering the fresh rBC emission and the "apparent" $D_p$ growth rate. And this true growth rate value is important in the atmospheric model. Unfortunately, the emission data about the aging degree of rBC from different emission sectors is still lacking. We may conduct laboratory experiments to determine the rBC aging degree from varied rBC sources and try to estimate the true $D_p$ growth rate in the future.

Line 305 - This sentence is worded as if the Li et al 2003 study took images of the rBC from this study, which is obviously not correct. Were any new microscopy images taken from the current study period?

Reply: There was no new microscopy images taken from current study. The literature cited here is to support the argument that bare rBC is in a fractal structure. We have changed the expression to make it clearer and added some new literatures. Now:

This significant discrepancy indicates bare rBC was in a fractal structure consistent with the previous research from electron microscopic image that bare rBC was in a fractal chain-like structure (Li et al., 2003;Adachi and Buseck, 2013;Wang et al., 2017). (line 285-286)

Line 325 - What ambient measurements? From Peng et al 2016? These effective densities are nothing like what you report in the previous section.

425    Reply: Yes, it's from Peng et al 2016, we cite this literature in order to support the argument that the rBC-containing particles tend to become more compact with the coating increasing. The effective densities are the parameters in Peng's literature to support this argument. However, we think this sentence may confuse the readers and thus we changed the expression in the manuscript. Now:

Different techniques have been used to explore the morphology of rBC-containing particles in ambient and laboratory
430    measurements (Zhang et al., 2008;Peng et al., 2016;Pagels et al., 2009). It is generally agreed that the morphology of rBC-containing particles will become more compact with the aging process or with increasing coating thickness. (line 310-312)

Line 363 - Did you find the large uncertainty? Or did Liu et al 2017? Discuss more.

435    Reply: Yes, both of our study and Liu found the uncertainty, typically in the external and transit stage because of the assumption in the morphology-dependent model. The section about the light absorption have been rephrased.

Line 366 - What is the "morphology dependent model"? I am very confused by the whole Section 4.2.2.

440    Reply: We rephrase this section and specifically describe the morphology-dependent model in the supplementary.

Reply: What we mean is the wet scavenging may be a more efficient removal mechanism for larger rBC-containing particles.
445    We have changed the expression. Thanks for reminding.

**Technical corrections**

Great thanks for the patient and careful comments about the technical corrections from the reviewer, we have corrected the
450    technical corrections pointed by the reviewer and carefully checked the manuscript again and again. Thanks again for the reviewer for improving this paper.

Line 44 - should be "into the atmosphere"
Line 45 - need a comma after condensation

455     Reply: Now: ==After being emitted into the atmosphere, BC particles tend to mix with other substances through coagulation, condensation, and other photochemical process, which significantly changes BC's cloud condensation nuclei activity as well as its light absorption ability (Liu et al., 2013;Bond and Bergstrom, 2006).== (line 41-44)

Line 83-84 - reword: the data are not analyzed in the discussion section, they are presented

460     Reply: Now: ==A tandem experiment combining a centrifugal particle mass analyzer (CPMA, Cambustion Ltd.) and a differential mobility analyzer (DMA, model 3085A, TSI Inc., USA) with a SP2 were performed during two typical cases, focusing on BC-containing particles' microphysical properties.== (line 76-78)

Line 86-87 - reword "incandescent signal emissions"

465     Reply: Now: ==After a rBC-containing particle crosses the beam, it is heated to incandesce by sequentially absorbing the laser power.== (line 93-95)

Line 87 - What is "this rule"?

Reply: What we mean is we would use rBC as the abbreviation in the following section. Now:

470     ==For the SP2, the mass concentration of BC was measured on the basis of incandescent signal emissions; therefore, refractory black carbon (rBC) was used.== (line 81-82)

Line 95 - Fig 1b, specifically

Reply: Thanks, we have changed. (line 89)

475

Line 104 - reword: the laser intensity is not constant by performing PSL calibrations

Reply: We use a YAG power index in the housekeeping file to support the stability of our SP2. Yes, it's not constant. We rephrase the expression, now:

==The calibration of the scattering channel and incandescence channel was also conducted after the observation, the calibration==

480     ==coefficient varied little (<3%) and the YAG power (laser intensity index recorded by SP2) fluctuated with 4.8±0.1 indicating the stable condition of SP2 during the observation period.== (line 119-121)

Line 148 - unified should be unity; low should be lower

Reply: Now:

485     ==For large particles, the SP2 detection efficiency was approximately unity and decreased gradually for smaller rBC particles (Fig. S3).== (line 123-124)

Line 151 - without should be outside

Reply: Now:

490 By extrapolating a lognormal function fit to the observed mass distribution, we found that rBC particles outside the detection range caused an ~15% underestimation of the rBC mass concentration. (line 126-128)

Line 156 - is intracavity a noun?

Line 164-165 - reword "description ... described"

495 Line 170 - the densities are not defined in the text nor in Table S1

Line 174 - add "respectively" to the sentence

Line 179 - reword

Reply: These sentences have been deleted in the method section in the new manuscript.

500 Line 205 – reword

Reply: Former: $O_3$ dominant pollution occurred at 1400 LST on June 2, with a maximum of 145 ppbv, reflecting high atmospheric oxidant levels and strong photochemistry during the observation.

Now: The maximum $O_3$ concentration appeared at 1400 LST on June 2 with a value of 145 ppbv, reflecting high atmospheric oxidant levels and strong photochemistry during the observation. (line 185)

505

Line 207 - Don't use "this" as the subject of a sentence.

Reply: Former:" The mass concentration of rBC was $1.21 \pm 0.73$ μg/m$^3$ on average, accounting for $3.5 \pm 2.4\%$ of PM$_{2.5}$ on an hourly basis. This was comparable to the previous filter-based measurement in Beijing, with an average fraction of 3.2% in the summer of 2010 (Zhang et al., 2013)."

510 Now: The mass concentration of rBC was $1.21 \pm 0.73$ μg/m$^3$ on average, accounting for $3.5 \pm 2.4\%$ of PM$_{2.5}$ on an hourly basis, which was comparable to the previous filter-based measurement in Beijing, with an average fraction of 3.2% in the summer of 2010 (Zhang et al., 2013). (line 186-188)

Line 209-210 - reword, I don't think this is actually a sentence

515 Reply: It has been deleted.

Line 232 - MED or MMD?

Reply: It's MED. Liu et al. (2014) only reported the average MED and standard deviation of MED in winter and summer but not MMD, but these MED value also show a summer-low-winter-high trend.

520

Line 242 - combination should be combined

Reply: Now:

The diurnal cycle reached a peak plateau between 0300–0700 LST and it decreased gradually in the afternoon, which was controlled by the combined effects of the development of a planetary boundary layer (PBL) variation and on-road rBC emissions. (line 213-215)

Line 243 – reword

[revised manuscript text omitted]

**Section 1 Calibration**

Figure S1: The calibration of the incandescence channel. The data of incandescence peak and rBC mass is fitted using a poly function (y = ax$^2$+bx+c). The coefficient of the poly function varied little (<2%) before and after the observation indicating the stability of the incandescence channel. The scatter of incandescence intensity caused 25% uncertainty, resulting in an uncertainty of the derived BC mass of 20%, which causes an uncertainty of mass equivalent diameter of ~6%.

1170

Figure S2: The calibration of the scattering channel. The calibration factor varied little (<3%) before and after the observation indicating the stability of the scattering channel. The calibration is done using PSLwith multiple sizes (203 nm, 240 nm 300 nm, 400 nm) before the observation. And the calibration is done only with PSL with 240 nm after the observation.

1175 Figure S3: The calibration curve for the detection efficiency of SP2. For rBC with diameter > 70 nm, the detection efficiency is larger than 80%.

Figure S4: The calibration of the DMA-SP2 system. An DMA-SP2 system can determine the effective density of rBC. We test our DMA-SP2 system by measuring the effective density of aquadag and comparing the result with previous research. Our
1180 results are ~7% higher than the poly-fit of Gysel but lower than the results from Moteki and Kondo. These differences may be result of different characteristics of Aquadag with varied lot and different instrument condition (such as the uncertainty of SP2).

[Figure]

**Figure S1 Calibration curve for incandescence broadband high gain channel.**

1185

[Figure]

**Figure S2 PSL calibration for high gain scattering channel before and after the investigation.**

[Figure]

1190    **Figure S3 Calibration curve for SP2's detection efficiency.**

[Figure]

**Figure S4 Relationship between effective density and mobility diameter of Aquadag measured in the present study and in previous studies.**

1195

**Section 2 Backward trajectory**

An WRF-Flexpart model (https://www.flexpart.eu) was used to analyze where the air mass was from. The 1°*1° FNL data
1200 (rda.ucar.edu/) was used as the input meteorological data to WRF. WRF can produce meteorological data with higher resolution which was used as the input data for Flexpart. Air samples were released at 100m above ground level at the observation site (longitude: 116.37°E; latitude: 39.97°N) and the simulation time of backward trajectory is 3 days.

On the clean days (06/07, 06/08) the air mass was from the north of Beijing where there is little pollutant emission. Since the north air mass is clean, the local emitted pollutant may be dominant.
1205 On the pollution day (06/12, 06/13), the air mass was majorly from the south polluted area. The pollutant transportation may paly an important role in pollution day.

[Figure]

**Figure S5 The backward trajectories at clean days (06/07, 06/08) and polluted days (06/12, 06/13), the map is the built-in map of the IGOR software (https://www.wavemetrics.com/).**

1210

**Section 3 Absorption enhancement calculation**

 ### 3.1 Description of morphology-dependent model

The absorption enhancement ($E_{ab}$) of a single rBC-containing particle is calculated as the ratio of absorption cross section of rBC-containing particle ($C_{abs,p}$) and absorption cross section of rBC core ($C_{abs,rBC}$) using Mie theory assuming a core-shell structure with refractive indices of 2.26+1.26i for rBC core and 1.48 for coatings.

$$E_{abs\_coreshell} = \frac{C_{abs,p}}{C_{abs,rBC}}$$

Considering rBC-containing particle is not in an ideal core-shell structure as discussed in section 4.1.2, the rBC-containing particles was classified into external, transition and core-shell stage based on the $M_R$ range. The rBC-containing particles with an external mixing state were considered to have no absorption enhancement, and the rBC-containing particles at the core-shell stage were considered to have a core-shell structure and the same $E_{abs}$ from Mie-theory under the assumption of a perfect core-shell structure. The $E_{abs}$ in the transition period was calculated by the interpolation of $E_{abs}$ between the external and internal stage, which can be explained as the following equation:

$$E_{abs\_new} = \begin{cases} 1 & when\ M_R \leq 1.5 \\ \frac{E_{abs\_coreshell}(M_R=6) - E_{abs\_coreshell}(M_R=1.5)}{6-1.5} * (M_R - 1.5) + 1 & when\ 1.5 < M_R < 6 \\ E_{abs\_coreshell} & when\ M_R \geq 6 \end{cases}$$

The reliability of this morphology- dependent model has been proven by comparing the $E_{abs}$ derived from the model and measuring the $E_{abs}$ (Liu et al., 2017).

[Figure]

1230

**Figure S6 Dependence of $E_{abs}$ on $M_R$ at wavelength of 550 nm and $D_c$ = 180 nm, calculated using the Mie model under the assumption of a core-shell structure (red solid line). The gray dashed line denotes the $E_{abs}$ calculated from morphology-dependent model.**

1235

**3.2 Applying the morphology-dependent model**

The $D_p$ and $D_c$ can be directly obtained in the single SP2 measurement. With the rBC density of 1.8 g/cm$^3$ and assuming a coating density of 1.5 g/cm$^3$, the $M_R$ of every single rBC-containing particle can be calculated in the ambient measurement. Thus, the relationship of the morphology-dependent model between $M_R$ and $E_{abs}$ can be used. We calculated the $E_{abs}$ of every single rBC-containing particle with $D_c$ =180 nm in one hour and reported the average $E_{abs}$ in Fig. 12.

1240

---

## Author Response (AR3)

**Reply to the comments of anonymous reviewer #1 on manuscript entitled " Mixing characteristics of refractory black carbon aerosols at an urban site in Beijing "**

The manuscript has clearly improved in terms of clarity, specially the methodology section through more detailed explanations of the techniques and the experiments performed. The English language has also been improved.

The study combines different approaches to apportion/characterize rBC particles and its mixing state in a measurement site near Beijing during summer, which I consider to be suitable for publication in ACP. However, in my opinion, there are several points that need improvement before final publication, especially in terms of presenting the results in a precise, logical sequence and integrating the different types of analysis. The results subsections appear to be decoupled from each other, which make it difficult to read. This concern could be improved, at first, by reformulating the last paragraph of the introduction (L75-L82) and briefly presenting the organization of the paper. This paragraph could "… provide the readers with the expectation of what they will find out by reading your paper" (Schultz, David M, 1965 – Eloquent Science). Moreover, I think the following aspects need further clarification:

Reply: Thanks for the confirmation of the reviewer for our new manuscript and the patient comments.

We have modified the description of the experiment and paper (L75-L85) to let the readers know the structure of this paper easier.

Line "75-76", the size distribution and coating thickness of BC-containing particles corresponds to the results part.

Line "76-79", the microphysical properties corresponds to the discussion part.

We added "The results of this paper were exhibited in the following sequence: (1) the size distribution of BC core and it's influence factors such as seasons, weather condition and etc.; (2) the coating thickness of BC and it's influence factors such as diurnal variation, $O_x$ condition and etc.; (3) the morphology of BC and it's relationship with coating thickness; (4) the relationship of morphology of BC with it's light absorption." (Line 79-83)

L78: Please specify what are the typical cases and the microphysical properties.

Reply: We change the expression "typical cases" to "the clean and polluted cases". Line 78

We have specified the microphysical properties in Line 79.

L212-214: You mention a clear diurnal pattern – is it in terms of concentration or MMD? I think the whole discussion (L212-215) could be better illustrated by a diurnal cycle plot. In my opinion, the type of plot does not correspond with the discussion.

Reply: We refer to the diurnal cycle of dM/dlog$D_c$. As shown in Fig. 5, the value of dM/dlog$D_c$ is higher at night with the red color and the value of dM/dlog$D_c$ is lower at noon with the blue color. It's dM/dlogD$_c$ but not the MMD and we think the diurnal pattern is clear in Fig. 5. We have specified that the diurnal cycle is for dM/dlogD$_c$ in the new manuscript. (Line 217).

L216-217: The same as the comment before. This sentence could be better illustrated by showing the size distributions of before vs. after rain, or similar. The statement cannot be easily observed in Fig 5 or in Fig.3.

Reply: We have added Figure S7 in the supplementary to illustrate the change of the size distribution before and after and rain cases and mentioned in Line 221.

[Figure]

**Figure S7 Size distribution of the rBC core before and after the typical rain cases.**

L228-230: Is the larger MMD in southerly air masses only due to the different combustion source or is it also related to the aging time in the atmosphere?

Reply: We think the size of BC core can't be influenced by the aging since BC is inertial and can't easily react with other substance. The aging time can influence the coating thickness of BC but can't influence the size of BC core. Thus, the MMD is not related to aging time in our opinion.

L246-252: In my opinion Figure 7 is quite complex and with some excess of information that is not used in the discussion or necessary to prove the main point. I suggest 'cleaning' this figure. Moreover, I could not fully understand the colormap (dN/dlogDp) vs. (Dp/Dc, in linear scale). Could you please clarify how did you calculate this? The same applies for the upper panels. I think the "dN/dlogDp" should be a "normalized frequency", please check. Also consider to give indexes (a), (b) and (c) for each of the panels in Fig. 7.

Reply: We have done some cleaning by removing the $PM_{2.5}$ information in this Figure since it can be found elsewhere and we have added more description of the figure (Line 570) and give indexes for each of the panels in Fig. 7.

The colormap is the frequency (dN) of the rBC-containing particles with fix $D_c$ and varied $D_p/D_c$ (vertical axes) and varied time (horizontal axes), the frequency (dN) is scaled by the $logD_p$ and denoted by $dN/dlogD_p$.

Figs 3, 5 and 7: please put the x-axis in the same time format and same time range since they all correspond to the same measurement. You should also consider to combine them into one single plot, with the information of the time window of the rain events, the tandem experiment and other events that were used in the analyses. Note that the PM2.5 is presented in all of them.

Reply: We think it's a good advice to put the x-axis in the same time format and we have changed the figures. The time range of Fig.3 is a bit bigger than Fig. 5 and Fig. 7 since we want to show the time of the second tandem system measurement and the condition (such as the concentration of $PM_{2.5}$ and weather) during the second tandem measurement.

We think the picture may be too big if we put all the time series together and we think it may be better to arrange them in the same sequence with writing.

L252-256: I found these sentences quite confusing.

Reply: We have rewritten this part. (Line 255-261)

L255: The authors say Dc/Dp = 1.8 in L255 but Dc/Dp = 1.4 in L248 for the polluted case.

Reply: $D_p/D_c=1.4$ is the average value during the polluted period and the $D_p/D_c=1.8$ is the right peak location of $D_p/D_c$ distribution during the polluted period.

L258-272: You consider the aging process responsible for the increase in Dp/Dc. Could that be related to the advection of aged particles (thickly coated) when the BL is developed? Did you consider discriminating the periods when the site is influenced by northerly or southerly winds?

Reply: Yes, the aged particles from the upper boundary layer may also explain the daily variation of $D_p/D_c$ and we added this suspect in Line 269-270. More work need to be done in the vertical measurement of $D_p/D_c$ to demonstrate this suspect.

It's true that the $D_p/D_c$ may be related to the wind direction because of the different sources and aging processes of rBC. We would like to explore more in the future.

Minor comments:

Figure 1. The paper is about influence of Beijing outflow on the measurement site. I suggest pointing the city on the map in
(a) or outline the city borders in (b). This will be very helpful for the reader.

Reply: We have decreased the size of the red box in Fig.1 (a) and the red box can exactly reflect the borders exhibited in Fig.1
(b).

Figure 4. You present the number size distribution in Fig. 4 but make no reference to that in the text.

Reply: We have added some description in Line 206-207, since we mainly focus on mass size distribution we don't want to
discuss more about number size distribution.

L22: The microphysical properties of rBC were also studied.

Reply: Thanks, it has been changed. (Line 22)

L94: intense vs. intensive.

Reply: Thanks, it has been changed. (Line 95)

L96: To be consistent, use "rBC mass".

Reply: Thanks, it has been changed. (Line 97)

L 106: Put the equation in a separate line and number it.
Reply: Thanks, it has been changed. (Line 110)

L 115: Liu et al. (2015)
When presenting your results, always round numbers to appropriate digits. (e.g. L186, L194, L201)
(https://en.wikipedia.org/wiki/Significant_figures).

Reply: Thanks, it has been changed. (Line 121, Line 191, Line 199, Line 207, Line 365). We have checked the problem
elsewhere and changed the expression. (Line 21, Line 29, Line 165, Line 251, Line 252, Line 266, Line 271, Line 278, Line
289, Line 301, Line 302, Line 361, Line 367, Line 382, Line 392)

L149: Please rephrase: "where many heavy industries are located". Moreover, I assume it is not only industries, correct?

130 Reply: Yes, it's one of the most polluted area in China and there are substantial emissions from industry, residential, biomass burning and etc. We have modified the expression (Line 154).

L225 and Fig. 5 – As I am not familiar with wind rose plots, I was puzzled when you made the reference to the angle values. Where does your angle scale starts? I would expect the angle of 45 degrees to be heading in a NE direction.

135

Reply: Yes, the angle of 45 degrees (Cartesian coordinate) represents the NE direction and the angle of 0 degree represents the E direction. This rule is same for all wind rose plots.

L239: investigation period.

140

Reply: Thanks, it has been changed. (Line 243)

L378: On "A simulation showed that the Eabsaveraged" there is a space missing.

145 Reply: Thanks, it has been changed. (Line 383)

L303: Maybe "mixture" instead of "structure".

Reply: We changed "structure" to "mixing structure". (Line 308)

150

L304: "approaches" instead of "was equal to".

Reply: Thanks, it has been changed. (Line 309)

155 Avoid starting sentences with symbols or acronyms (e.g. L24, L260, L282, L297, L346).

Reply: Thanks a lot, we have changed the expression to avoid such problems. (Line 24, Line 39, Line 42, Line 270, Line 292, Line 307, Line 356). We feel sorry that we misunderstand the reviewer's suggestion in the last reply. We have carefully checked such problem in the paper and modified such problem. (Line 249, Line 374)

[revised manuscript text omitted]

**Section 1 Calibration**

Figure S1: The calibration of the incandescence channel. The data of incandescence peak and rBC mass is fitted using a poly function ($y = ax^2+bx+c$). The coefficient of the poly function varied little (<2%) before and after the observation indicating the stability of the incandescence channel. The scatter of incandescence intensity caused 25% uncertainty, resulting in an uncertainty of the derived BC mass of 20%, which causes an uncertainty of mass equivalent diameter of ~6%.

Figure S2: The calibration of the scattering channel. The calibration factor varied little (<3%) before and after the observation indicating the stability of the scattering channel. The calibration is done using PSLwith multiple sizes (203 nm, 240 nm 300 nm, 400 nm) before the observation. And the calibration is done only with PSL with 240 nm after the observation.

Figure S3: The calibration curve for the detection efficiency of SP2. For rBC with diameter > 70 nm, the detection efficiency is larger than 80%.

Figure S4: The calibration of the DMA-SP2 system. An DMA-SP2 system can determine the effective density of rBC. We test our DMA-SP2 system by measuring the effective density of aquadag and comparing the result with previous research. Our results are ~7% higher than the poly-fit of Gysel but lower than the results from Moteki and Kondo. These differences may be result of different characteristics of Aquadag with varied lot and different instrument condition (such as the uncertainty of SP2).

[Figure]

**Figure S1 Calibration curve for incandescence broadband high gain channel.**

[Figure]

**Figure S2 PSL calibration for high gain scattering channel before and after the investigation.**

[Figure]

800    **Figure S3 Calibration curve for SP2's detection efficiency.**

[Figure]

**Figure S4 Relationship between effective density and mobility diameter of Aquadag measured in the present study and in previous studies.**

805

**Section 2 Backward trajectory**

An WRF-Flexpart model (https://www.flexpart.eu) was used to analyze where the air mass was from. The 1 °*1 °FNL data (rda.ucar.edu/) was used as the input meteorological data to WRF. WRF can produce meteorological data with higher resolution which was used as the input data for Flexpart. Air samples were released at 100m above ground level at the

810    observation site (longitude: 116.37 °E; latitude: 39.97 °N) and the simulation time of backward trajectory is 3 days.

On the clean days (06/07, 06/08) the air mass was from the north of Beijing where there is little pollutant emission. Since the north air mass is clean, the local emitted pollutant may be dominant.

On the pollution day (06/12, 06/13), the air mass was majorly from the south polluted area. The pollutant transportation may play an important role in pollution day.

815

[Figure]

**Figure S5 The backward trajectories at clean days (06/07, 06/08) and polluted days (06/12, 06/13), the map is extracted from Igor Pro software (© 2016 wavemetrics, www.wavemetrics.com).**

**Section 3 Absorption enhancement calculation**

**3.1 Description of morphology-dependent model**

The absorption enhancement ($E_{ab}$) of a single rBC-containing particle is calculated as the ratio of absorption cross section of rBC-containing particle ($C_{abs,p}$) and absorption cross section of rBC core ($C_{abs,rBC}$) using Mie theory assuming a core-shell structure with refractive indices of 2.26+1.26i for rBC core and 1.48 for coatings.

$$E_{abs\_coreshell} = \frac{C_{abs,p}}{C_{abs,rBC}}$$

Considering rBC-containing particle is not in an ideal core-shell structure as discussed in section 4.1.2, the rBC-containing particles was classified into external, transition and core-shell stage based on the $M_R$ range. The rBC-containing particles with an external mixing state were considered to have no absorption enhancement, and the rBC-containing particles at the core-shell stage were considered to have a core-shell structure and the same $E_{abs}$ from Mie-theory under the assumption of a perfect core-shell structure. The $E_{abs}$ in the transition period was calculated by the interpolation of $E_{abs}$ between the external and internal stage, which can be explained as the following equation:

$$E_{abs\_new} = \begin{cases} 1 & when\ M_R \leq 1.5 \\ \frac{E_{abs\_coreshell}(M_R=6) - E_{abs\_coreshell}(M_R=1.5)}{6-1.5} * (M_R - 1.5) + 1 & when\ 1.5 < M_R < 6 \\ E_{abs\_coreshell} & when\ M_R \geq 6 \end{cases}$$

The reliability of this morphology- dependent model has been proven by comparing the $E_{abs}$ derived from the model and measuring the $E_{abs}$ (Liu et al., 2017).

[Figure]

**Figure S6 Dependence of Eabs on MR at wavelength of 550 nm and $D_c$ = 180 nm, calculated using the Mie model under the assumption of a core-shell structure (red solid line). The gray dashed line denotes the Eabs calculated from morphology-dependent model.**

**3.2 Applying the morphology-dependent model**

The $D_p$ and $D_c$ can be directly obtained in the single SP2 measurement. With the rBC density of 1.8 g/cm$^3$ and assuming a coating density of 1.5 g/cm$^3$, the $M_R$ of every single rBC-containing particle can be calculated in the ambient measurement. Thus, the relationship of the morphology-dependent model between $M_R$ and $E_{abs}$ can be used. We calculated the $E_{abs}$ of every single rBC-containing particle with $D_c$ =180 nm in one hour and reported the average $E_{abs}$ in Fig. 12.

[Figure]

850 **Figure S7 Size distribution of the rBC core before and after the typical rain cases.**